# Self-Verifying Reflection
# Helps Transformers with CoT Reasoning

**Zhongwei Yu[1], Wannian Xia[2], Xue Yan[2], Bo Xu[2], Haifeng Zhang[2], Yali Du[3], Jun Wang[4]**

[1] The Hong Kong University of Science and Technology (Guangzhou)
[2] Institute of Automation, Chinese Academy of Sciences
[3] King's College London [4] University College London
[1] `zyu950@connect.hkust-gz.edu.cn,`
[2] `{xiawannian2020, yanxue2021, xubo, haifeng.zhang}@ia.ac.cn,`
[3] `yali.du@kcl.ac.uk,` [4] `jun.wang@cs.ucl.ac.uk`

## Abstract

Advanced large language models (LLMs) frequently reflect in reasoning chain-of-thoughts (CoTs), where they self-verify the correctness of current solutions and explore alternatives. However, given recent findings that LLMs detect limited errors in CoTs, how reflection contributes to empirical improvements remains unclear. To analyze this issue, in this paper, we present a minimalistic reasoning framework to support basic self-verifying reflection for small transformers without natural language, which ensures analytic clarity and reduces the cost of comprehensive experiments. Theoretically, we prove that self-verifying reflection guarantees improvements if verification errors are properly bounded. Experimentally, we show that tiny transformers, with only a few million parameters, benefit from self-verification in both training and reflective execution, reaching remarkable LLM-level performance in integer multiplication and Sudoku. Similar to LLM results, we find that reinforcement learning (RL) improves in-distribution performance and incentivizes frequent reflection for tiny transformers, yet RL mainly optimizes shallow statistical patterns without faithfully reducing verification errors. In conclusion, integrating generative transformers with discriminative verification inherently facilitates CoT reasoning, regardless of scaling and natural language.

## 1 Introduction

Numerous studies have explored the ability of large language models (LLMs) to reason through a chain of thought (CoT), an intermediate sequence leading to the final answer. While simple prompts can elicit CoT reasoning [13], subsequent works have further enhanced CoT quality through reflective thinking [10] and the use of verifiers [4]. Recently, reinforcement learning (RL) [33] has achieved notable success in advanced reasoning models, such as OpenAI-o1 [20] and Deepseek-R1 [5], which show frequent reflective behaviors that self-verify the correctness of current solutions and explore alternatives, integrating generative processes with discriminative inference. However, researchers also report that the ability of these LLMs to detect errors is rather limited, and a large portion of reflection fails to bring correct solutions [11]. Given the weak verification ability, the experimental benefits of reflection and the emergence of high reflection frequency in RL require further explanation.

To address this challenge, we seek to analyze two main questions in this paper: 1) what role self-verifying reflection plays in training and execution of reasoning models, and 2) how reflective reasoning evolves in RL with verifiable outcome rewards [15]. However, the complexity of natural language and the prohibitive training cost of LLMs make it difficult to draw clear conclusions from theoretical abstraction and comprehensive experiments across settings. Inspired by Zeyuan *et al.*

39th Conference on Neural Information Processing Systems (NeurIPS 2025).

[2], we observe that task-specific reasoning and self-verifying reflection do not necessitate complex language. This allows us to investigate reflective reasoning through tiny transformer models [36], which provide efficient tools to understand self-verifying reflection through massive experiments.

To enable tiny transformers to produce long reflective CoTs and ensure analytic simplicity, we introduce a minimalistic reasoning framework, which supports essential reasoning behaviors that are operable without natural language. In our study, the model self-verifies the correctness of each thought step; then, it may resample incorrect steps or trace back to previous steps. Based on this framework, we theoretically prove that self-verifying reflection improves reasoning accuracy if verification errors are properly bounded, which does not necessitate a strong verifier. Additionally, a trace-back mechanism that allows revisiting previous solutions conditionally improves performance if the problem requires a sufficiently large number of steps.

Our experiments evaluate 1M, 4M, and 16M transformers in solving integer multiplication [7] and Sudoku puzzles [3], which have simple definitions (thus, operable by transformers without language) yet still challenging for even LLM solvers. To maintain relevance to broader LLM research, the tiny transformers are trained from scratch through a pipeline similar to that of training LLM reasoners. Our main findings are listed as follows: 1) Learning to self-verify greatly facilitates the learning of forward reasoning. 2) Reflection improves reasoning accuracy if true correct steps are not excessively verified as incorrect. 3) Resembling the results of DeepSeek-R1 [5], RL can incentivize reflection if the reasoner can effectively explore potential solutions. 4) However, RL fine-tuning increases performance mainly statistically, with limited improvements in generalizable problem-solving skills.

Overall, this paper contributes to the fundamental understanding of reflection in reasoning models by clarifying its effectiveness and synergy with RL. Our findings based on minimal reasoners imply a general benefit of reflection for more advanced models, which operate on a super-set of our simplified reasoning behaviors. In addition, our implementation also provides insights into the development of computationally efficient reasoning models.

## 2   Related works

**CoT reasoning**   Pretrained LLMs emerge the ability to produce CoTs from simple prompts [13, 38], which can be explained via the local dependencies [25] and probabilistic distribution [35] of natural-language reasoning. Many recent studies develop models targeted at reasoning, e.g., scaling test-time inference with external verifiers [4, 17, 18, 32] and distilling large general models to smaller specialized models [34, 9]. In this paper, we train tiny transformers from scratch to not only generate CoTs but also self-verify, i.e., detect errors in their own thoughts without external models.

**RL fine-tuning for CoT reasoning**   RL [33] recently emerges as a key method for CoT reasoning [31, 40]. It optimizes the transformer model by favoring CoTs that yield high cumulated rewards, where PPO [29] and its variant GRPO [31] are two representative approaches. Central to RL fine-tuning are reward models that guide policy optimization: the 1) *outcome reward models* (ORM) assessing final answers, and the 2) *process reward models* (PRM) [17] evaluating intermediate reasoning steps. Recent advances in *RL with verifiable rewards* (RLVR) [5, 41] demonstrate that simple ORM based solely on answer correctness can induce sophisticated reasoning behaviors.

**Reflection in LLM reasoning**   LLM reflection provides feedback to the generated solutions [19] and may accordingly refine the solutions [10]. Research shows that supervised learning from verbal reflection improves performance, even though the reflective feedback is omitted during execution [42]. Compared to the generative verbal reflection, self-verification uses discriminative labels to indicate the correctness of reasoning steps, which supports reflective execution and is operable without linguistic knowledge. Recently, RL is widely used to develop strong reflective abilities [14, 27, 20]. In particular, DeepSeek-R1 [5] shows that RLVR elicits frequent reflection, and such a result is reproduced in smaller LLMs [24]. In this paper, we further investigate how reflection evolves during RLVR by examining the change of verification errors.

**Understanding LLMs through small transformers**   Small transformers are helpful tools to understand LLMs, for their architectural consistency with LLMs and low development cost to support massive experiments. For example, transformers smaller than 1B provide insights into how data mixture and data diversity influence LLM training [39, 2]. They also contribute to foundational

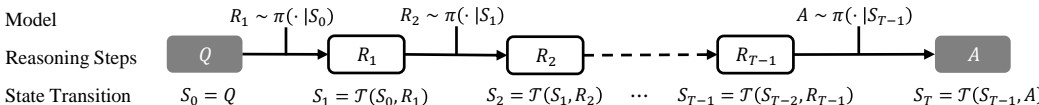

Figure 1: The illustration of MTP, where the transformer model $\pi$ reasons the answer $A$ of a query $Q$ through $T-1$ intermediate steps.

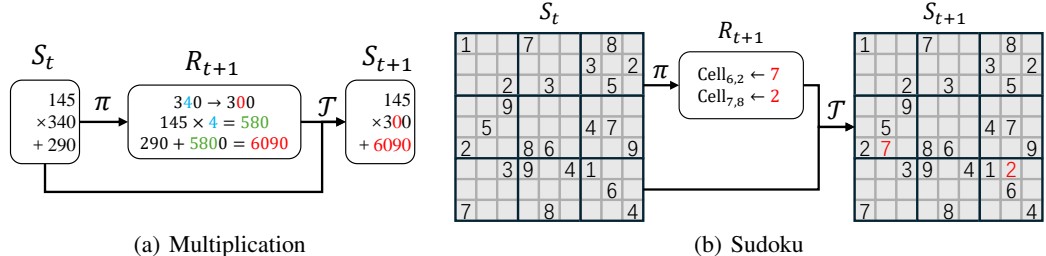

| (a) Multiplication | (b) Sudoku |

Figure 2: Example reasoning steps for multiplication and Sudoku, where the core planning is presented in the reasoning step $R_{t+1}$.

understanding of CoT reasoning, such as length generalization [12], internalization of thoughts [6], and how CoTs inherently extend the problem-solving ability [8, 16]. In this paper, we further use tiny transformers to better understand reflection in CoT reasoning.

## 3 Reflective reasoning for transformers

In this section, we develop transformers to perform simple reflective reasoning in long CoTs. Focusing on analytic clarity and broader implications, the design of our framework follows the minimalistic principle, providing only essential reasoning behavior operable without linguistic knowledge. More advanced reasoning frameworks optimized for small-scale models are certainly our next move in future work. In the following, we first introduce the basic formulation of CoT reasoning; then, based on this formulation, we introduce our simple reasoning framework for self-verifying reflection; afterwards, we describe how transformers are trained to reason through this framework.

### 3.1 Reasoning formulation

**CoT Reasoning as a Markov decision process** A general form of CoT reasoning is given as a tuple $(Q, \{R\}, A)$, where $Q$ is the input query, $\{R\} = (R_1, \ldots, R_{T-1})$ is the sequence of $T-1$ intermediate steps, and $A$ is the final answer. Following Wang [37], we formulate the CoT reasoning as a *Markov thought process* (MTP). As shown in Figure 1, an MTP follows that [37]:

$$R_{t+1} \sim \pi(\cdot \mid S_t), \ S_{t+1} = \mathcal{T}(S_t, R_{t+1}), \tag{1}$$

where $S_t$ is the $t$-th reasoning state, $\pi$ is the *planning policy* (the transformer model), and $\mathcal{T}$ is the (usually deterministic) *transition function*. The initial state $S_0 := Q$ is given by the input query. In each reasoning step $R_{t+1}$, the policy $\pi$ plans the next reasoning action that determines the state transition, which is then executed by $\mathcal{T}$ to obtain the next state. The process terminates when the step presents the answer, i.e., $A = R_T$. For clarity, a **table of notations** is presented in Appendix A.

An MTP is implemented by specifying the state representations and transition function $\mathcal{T}$. Since we use tiny transformers that are weak in inferring long contexts, we suggest reducing the length of state representations, so that each state $S_t$ carries only necessary information for subsequent reasoning. Here, we present two examples to better illustrate how MTPs are designed for tiny transformers.

**Example 1** (An MTP for integer multiplication). As shown in Figure 2(a), to reason the product of two integers $x, y \geq 0$, each state is an expression $S_t := [x_t \times y_t + z_t]$ mathematically equal to $x \times y$, initialized as $S_0 = [x \times y + 0]$. On each step, $\pi$ plans $y_{t+1}$ by eliminate a non-zero digit in $y_t$ to 0, and it then computes $z_{t+1} = z_t + x_t(y_t - y_{t+1})$. Consequently, $\mathcal{T}$ updates $S_{t+1}$ as $[x_{t+1} \times y_{t+1} + z_{t+1}]$ with $x_{t+1} = x_t$. Similarly, $\pi$ may also eliminate non-zero digits in $x_t$ in a symmetric manner. Finally, $\pi$ yields $A = z_t$ as the answer if either $x_t$ or $y_t$ becomes 0.

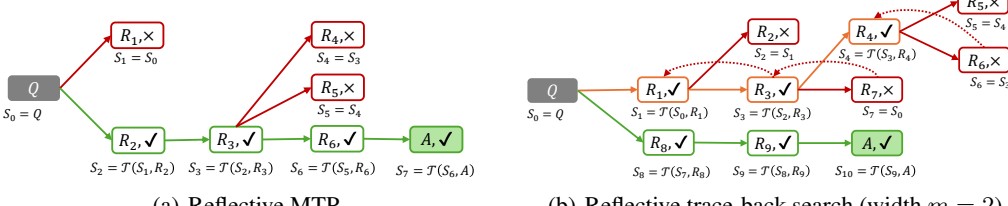

(a) Reflective MTP        (b) Reflective trace-back search (width $m = 2$)

Figure 3: Reflective reasoning based on MTP. "✓" and "×" are self-verification labels for positive and negative steps, respectively. The steps that are instantly verified as negative are highlighted in red. In RTBS, the dashed-line arrows back-propagate the negative labels, causing parental steps to be recursively rejected (orange). The green shows the steps that successfully lead to the answer.

**Example 2** (An MTP for Sudoku [3]). As shown in Figure 2(b), each Sudoku state is a $9 \times 9$ game board. On each step, the model $\pi$ fills some blank cells to produce a new board, which is exactly the next state. The answer $A$ is a board with no blank cells.

### 3.2  The framework of self-verifying reflection

Conceptually, reflection provides feedback for the proposed steps and may alter the subsequent reasoning accordingly. Reflection takes flexible forms in natural language (e.g., justifications and comprehensive evaluations), making it extremely costly to analyze. In this work, we propose to equip transformers with the simplest discriminative form of reflection, where the model self-verifies the correctness of each step and is allowed to retry those incorrect attempts. We currently do not consider the high-level revisory behavior that maps incorrect steps to correct ones, as we find learning such a mapping is challenging for tiny models and leads to no significant gain in practice. Specifically, we analyze two basic variants of reflective reasoning in this paper: the *reflective MTP* and the *reflective trace-back search*, as described below (see pseudo-code in Appendix D.1).

**Reflective MTP (RMTP)** Given any MTP with a policy $\pi$ and transition $\mathcal{T}$, we use a *verifier* $\mathcal{V}$ to produce a verification sequence after each reasoning step, denoted as $V_t \sim \mathcal{V}(\cdot|R_t)$. Such $V_t$ includes verification label(s): The positive "✓" and negative "×" signifying correct and incorrect reasoning of $R_t$, respectively. Given the *verified step* $\tilde{R}_{t+1} := (R_{t+1}, V_{t+1})$ that contains verification, we define $\tilde{\mathcal{T}}$ as the *reflective transition function* that rejects incorrect steps:

$$S_{t+1} = \tilde{\mathcal{T}}(S_t, \tilde{R}_{t+1}) = \tilde{\mathcal{T}}(S_t, (R_{t+1}, V_{t+1})) := \begin{cases} S_t, & \text{"×"} \in V_{t+1}; \\ \mathcal{T}(S_t, R_{t+1}), & \text{otherwise.} \end{cases} \quad (2)$$

In other words, if $\mathcal{V}$ detects any error (i.e. "×") in $R_{t+1}$, the state remains unchanged so that $\pi$ may re-sample another attempt. Focusing on self-verification, we use a single model called the *self-verifying policy* $\tilde{\pi} := \{\pi, \mathcal{V}\}$ to serve simultaneously as the *planning policy* $\pi$ and the verifier $\mathcal{V}$. By operating tokens, $\tilde{\pi}$ outputs the verified step $\tilde{R}_t$ for each input state $S_t$. In this way, $\tilde{\mathcal{T}}$ and $\tilde{\pi}$ constitute a new MTP called the RMTP, with illustration in Figure 3(a).

**Reflective trace-back search (RTBS)** Though RMTP allows instant rejections of incorrect steps, sometimes the quality of a step can be better determined by actually trying it. For example, a Sudoku solver occasionally makes tentative guesses and traces back if the subsequent reasoning fails. Inspired by o1-journey [26], a *trace-back search* allowing the reasoner to revisit previous states may be applied to explore solution paths in an MTP. We implement simple RTBS by simulating the depth-first search in the trajectory space. Let $m$ denote the *RTBS width*, i.e., the maximal number of attempts on each step. As illustrated in Figure 3(b), if $m$ proposed steps are rejected on a state $S_t$, the negative label "×" will be propagated back to *recursively reject* the previous step $R_t$. As a result, the state traces back to the closest ancestral state that has remaining attempt opportunities.

### 3.3  Training

As shown in Figure 4, we train the tiny transformers from scratch through consistent techniques of LLM counterparts, such as pretraining, supervised fine-tuning (SFT), and RL fine-tuning. First, we

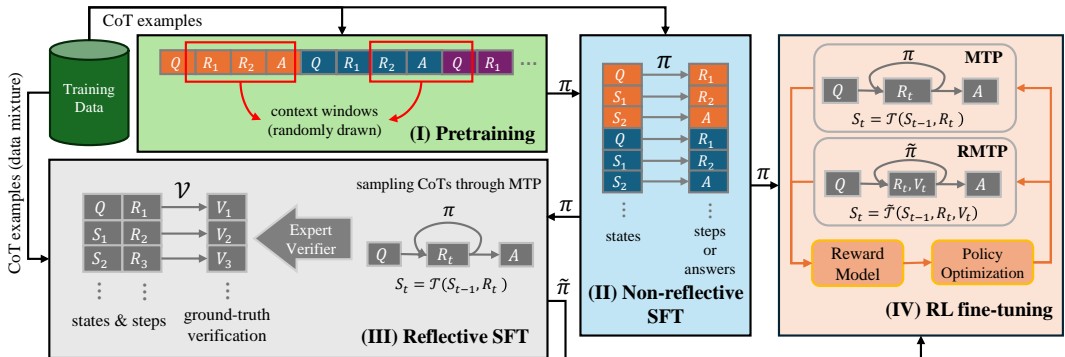

Figure 4: The training workflow for transformers to perform CoT reasoning.

use conventional pipelines to train a baseline model $\pi$ with only the planning ability in MTPs. During **(I) pretraining**, these CoT examples are treated as a textual corpus, where sequences are randomly drawn to minimize cross-entropy loss of next-token prediction. Then, in **(II) non-reflective SFT**, the model learns to map each state $S_t$ to the corresponding step $R_{t+1}$ by imitating examples.

Next, we employ **(III) reflective SFT** to integrate the planning policy $\pi$ with the knowledge of self-verification. To produce ground-truth verification labels, we use $\pi$ to sample non-reflective CoTs, in which the sampled steps are then labeled by an expert verifier (e.g., a rule-based process reward model). Reflective SFT learns to predict these labels from the states and the proposed steps, i.e., $(S_t, R_{t+1}) \rightarrow V_{t+1}$. To prevent disastrous forgetting, we also mix the *same* CoT examples as in non-reflective SFT. This converts $\pi$ to a self-verifying policy $\tilde{\pi}$ that can self-verify reasoning steps.

Thus far, we have obtained the planning policy $\pi$ and the self-verifying policy $\tilde{\pi}$, which can be further strengthened through **(IV) RL fine-tuning**. As illustrated in Figure 4, RL fine-tuning involves iteratively executing $\pi$ ($\tilde{\pi}$) to collect experience CoTs through an MTP (RMTP), evaluating these CoTs with a reward model, and updating the policy to favor higher-reward solutions. Following the RLVR paradigm [15], we use binary outcome rewards (i.e., 1 for correct answers and 0 otherwise) computed by a rule-based answer checker $\mathrm{ORM}(Q, A)$. When training the self-verifying policy $\tilde{\pi}$, the RMTP treats verification $V_t$ as a part of the augmented step $\tilde{R}_t$, simulating R1-like training [5] where reflection and solution planning are jointly optimized. We mainly use GRPO [31] as the algorithms to optimize policies. Details of RL fine-tuning are elaborated in Appendix B.

# 4 Theoretical results

This section establishes theoretical conditions under which self-verifying reflection (RMTP or RTBS in Section 3.2) enhances reasoning accuracy (the probability of deriving correct answers). The general relationship between the verification ability and reasoning accuracy (discussed in Appendix C.1) for any MTP is intractable as the states and transitions can be arbitrarily specified. Therefore, to derive interpretable insights, we discuss a simplified prototype of reasoning that epitomizes the representative principle of CoTs — to incrementally express complex relations by chaining the local relation in each step [25]. Specifically, Given query $Q$ as the initial state, we view a CoT as the step-by-step process that reduces the complexity within states:

- We define $\mathcal{S}_n$ as the set of states with a complexity scale of $n$. For simplicity, we assume that each step, if not rejected by reflection, reduces the complexity scale by 1. Therefore, the scale $n$ is the number of effective steps required to derive an answer.

- An answer $A$ is a state with a scale of 0, i.e. $A \in \mathcal{S}_0$. Given an input query $Q$, the answers $\mathcal{S}_0$ are divided into positive (correct) answers $\mathcal{S}_0^+$ and negative (wrong) answers $\mathcal{S}_0^-$.

- States $\mathcal{S}_n$ ($n > 0$) are divided into 1) *positive* states $\mathcal{S}_n^+$ that potentially lead to correct answers and 2) *negative* states $\mathcal{S}_n^-$ leading to only incorrect answers through forward transitions.

Consider a self-verifying policy $\tilde{\pi} = \{\pi, \mathcal{V}\}$ to solve this simplified task. We describe its fundamental abilities using the following probabilities (whose meanings will be explained afterwards):

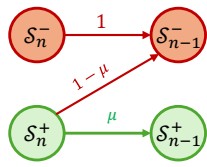

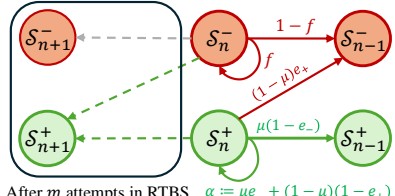

After $m$ attempts in RTBS $\quad \alpha := \mu e_- + (1-\mu)(1-e_+)$

(a) Non-reflective reasoning    (b) Reflective reasoning through an RMTP or RTBS

Figure 5: The diagram of state transitions starting from scale $n$ in the simplified reasoning, where probabilities are attached to solid lines. In (b) reflective reasoning, the dashed-line arrow presents the trace-back move after $m$ attempts in RTBS.

$$\mu := p_{R\sim\pi}(\mathcal{T}(S,R) \in \mathcal{S}_{n-1}^+ \mid S \in \mathcal{S}_n^+) \tag{3}$$

$$e_+ := p_{R,V\sim\tilde{\pi}}(\mathcal{T}(S,R) \in \mathcal{S}_{n-1}^-, \text{``}\times\text{''} \notin V \mid S \in \mathcal{S}_n^+), \tag{4}$$

$$e_- := p_{R,V\sim\tilde{\pi}}(\mathcal{T}(S,R) \in \mathcal{S}_{n-1}^+, \text{``}\times\text{''} \in V \mid S \in \mathcal{S}_n^+), \tag{5}$$

$$f := p_{R,V\sim\tilde{\pi}}(\text{``}\times\text{''} \in V \mid S \in \mathcal{S}_n^-). \tag{6}$$

To elaborate, $\mu$ measures the planning ability, defined as the probability that $\pi$ plans a step that leads to a positive next state, given that the current state is positive. For verification abilities, we measure the rates of two types of errors: $e_+$ (false positive rate) is the probability of accepting a step that leads to a negative state, and $e_-$ (false negative rate) is the probability of rejecting a step that leads to a positive state. Additionally, $f$ is the probability of rejecting any step on negative states, providing the chance of tracing back to previous states. Given these factors, Figure 5 illustrates the state transitions in non-reflective (vanilla MTP) and reflective (RMTB and RTBS) reasoning.

For input problems with scale $n$, we use $\rho(n)$, $\tilde{\rho}(n)$, and $\tilde{\rho}_m(n)$ to respectively denote the reasoning accuracy using no reflection, RMTP, and RTBS (with width $m$). Obviously, we have $\rho(n) = \mu^n$. In contrast, the mathematical forms of $\tilde{\rho}(n)$ and $\tilde{\rho}_m(n)$ are more complicated and therefore left to Appendix C.2. Our main result provides simple conditions for the above factors $(\mu, e_-, e_+, f)$ to ensure an improved accuracy when reasoning through an RMTP or RTBS.

**Theorem 1.** *In the above simplified problem, consider a self-verifying policy $\tilde{\pi}$ where $\mu$, $e_-$, and $e_+$ are non-trivial (i.e. neither 0 nor 1). Let $\alpha := \mu e_- + (1-\mu)(1-e_+)$ denote the rejection probability on positive states. Given an infinite computation budget, for $n > 0$ we have:*
- *$\tilde{\rho}(n) \geq \rho(n)$ if and only if $e_- + e_+ \leq 1$, where equalities hold simultaneously; furthermore, reducing either $e_-$ or $e_+$ strictly increases $\tilde{\rho}(n)$.*
- *$\tilde{\rho}_m(n) > \tilde{\rho}(n)$ for a sufficiently large $n$ if and only if $f > \alpha$ and $m > \frac{1}{1-\alpha}$; furthermore, such a gap of $\tilde{\rho}_m(n)$ over $\tilde{\rho}(n)$ increases strictly with $f$.*

**Does reflection require a strong verifier?** Theorem 1 shows that RMTP improves performance over vanilla MTP if the verification errors $e_+$ and $e_-$ are properly bounded, which *does not necessitate a strong verifier*. In our simplified setting, this only requires the verifier $\mathcal{V}$ to be better than random guessing (which ensures $e_- + e_+ = 1$). This also indicates a trivial guarantee of RTBS, as an infinitely large width ($m \to +\infty$) substantially converts RTBS to RMTB.

**When does trace-back search facilitate reflection?** Theorem 1 provides the conditions for RTBS to outperform RMTP for a *sufficiently large $n$*: 1) The width $m$ is large enough to ensure *effective exploration*. 2) $f > \alpha$ indicates that *negative states are inherently discriminated* from positive ones, leading to a higher rejection probability on negative states than on positive states (see Figure 5(b)). In other words, provided $f > \alpha$, RTBS is ensured to be more effective on complicated queries using a finite $m$. However, this also implies a risk of over-thought on simple queries that have a small $n$.

The derivation and additional details of Theorem 1 are provided in Appendix C.3. In addition, we also derive how many steps it costs to find a correct solution in RMTP. The following Proposition 1 (see proof in Appendix C.4) shows that a higher $e_-$ causes more steps to be necessarily rejected and increases the solution cost. In contrast, although a higher $e_+$ reduces accuracy, it forces successful solutions to rely less on reflection, leading to fewer expected steps. Therefore, *a high false negative rate $e_-$ is worse than a high $e_+$* given the limited computational budget in practice.

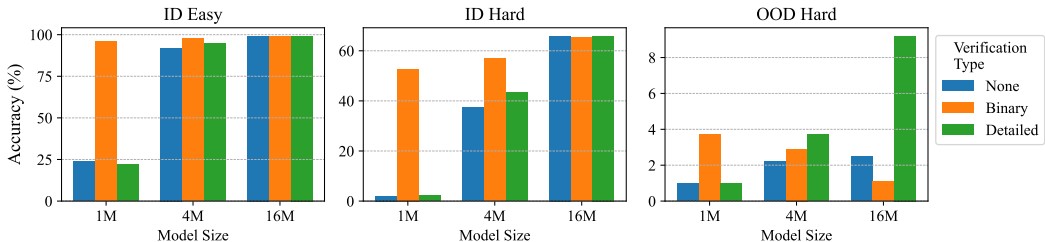

Figure 6: The accuracy of *non-reflective execution* of models in Mult. In each group, we compare training with various types of verification ("None" for no reflective SFT).

**Proposition 1** (RMTP Reasoning Length). *For a simplified reasoning problem with scale $n$, the expected number of steps $\bar{T}$ for $\tilde{\pi}$ to find a correct answer is $\bar{T} = \frac{n}{(1-\mu)e_+ + \mu(1-e_-)}$. Especially, a correct answer will never be found if the denominator is $0$.*

Appendix C.5 further extends our analysis to more realistic reasoning, where rejected attempts lead to a posterior drop of $\mu$ (or rise of $e_-$), indicating that the model may not well generalize the current state. In this case, the bound of $e_-$ to ensure improvements becomes stricter than that in Theorem 1.

# 5 Experiments

We conduct comprehensive experiments to examine the reasoning performance of tiny transformers under various settings. We trained simple causal-attention transformers [36] (implemented by LitGPT [1]) with 1M, 4M, and 16M parameters, through the pipelines described in Section 3.3. Details of training data, model architectures, tokenization, and hyperparameters are included in Appendix D. The source code is available at `https://github.com/zwyu-ai/self-verifying-reflection-reasoning`.

We test tiny transformers in two reasoning tasks: The integer multiplication task (**Mult** for short) computes the product of two integers $x$ and $y$; the **Sudoku** task fills numbers into blank positions of a $9 \times 9$ matrix, such that each row, column, or $3 \times 3$ block is a permutation of $\{1, \ldots, 9\}$. For both tasks, we divide queries into *3 levels of difficulties*: The **in-distribution (ID) Easy**, **ID Hard**, and **out-of-distribution (OOD) Hard**. *The models are trained on ID-Easy and ID-Hard problems*, while tested additionally on OOD-Hard cases. We define the difficulty of a Mult query by the number $d$ of digits of the greater multiplicand, and that of a Sudoku puzzle is determined by the number $b$ of blanks to be filled. Specifically, we have $1 \leq d \leq 5$ or $9 \leq b < 36$ for ID Easy, $6 \leq d \leq 8$ or $36 \leq b < 54$ for ID Hard, and $9 \leq d \leq 10$ or $54 \leq b < 63$ for OOD Hard.

Our full results are presented in Appendix E. Shown in Appendix E.1, these seemingly simple tasks pose challenges even for some well-known LLMs. Remarkably, through simple self-verifying reflection, our best 4M Sudoku model is as good as OpenAI o3-mini [21], and our best 16M Mult model outperforms DeepSeek-R1 [5] in ID difficulties.

## 5.1 Results of supervised fine-tuning

First, we conduct (I) pretraining, (II) non-reflective SFT, and (III) reflective SFT as described in Section 3.3. In reflective SFT, we consider learning two **types of self-verification**: 1) The **binary verification** includes a single binary label indicating the overall correctness of a planned step; 2) the **detailed verification** includes a series of binary labels checking the correctness of each meaningful element in the step. The implementation of verification labels is elaborated in Appendix D.2.3. We present our full SFT results in Appendix E.2, which includes training 30 models and executing 54 tests. In the following, we discuss our main findings through visualizing representative results.

**Does learning self-verification facilitate learning the planning policy?** We compare our models under the *non-reflective execution*, where self-verification is not actively used in test time. As shown in Figure 6, reflective SFT with *binary verification* brings remarkable improvements for 1M and 4M in ID-Easy and ID-Hard Mult problems, greatly reducing the gap among model sizes. Although detailed verification does not benefit as much as binary verification in ID problems, it significantly benefits the 16M model in solving OOD-Hard problems. Therefore, *learning to self-verify benefits the learning of forward planning, increasing performance even if test-time reflection is not enabled.*

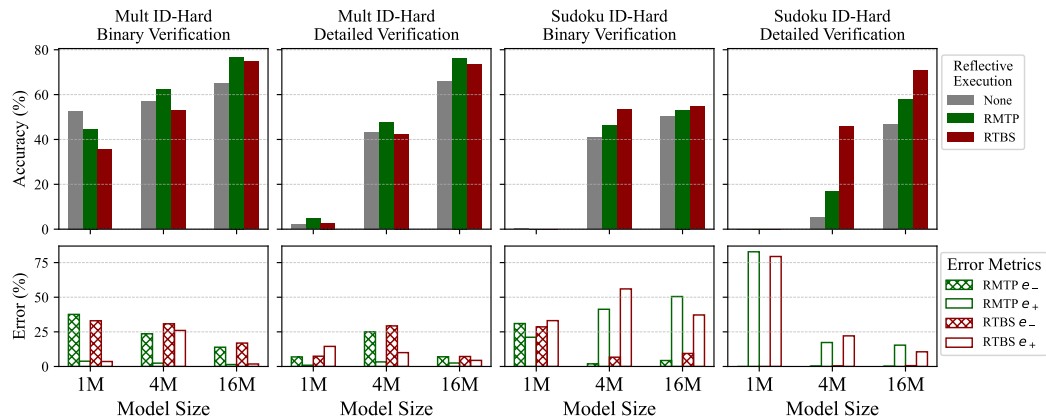

Figure 7: Performance of reflective execution methods across different model sizes, including the accuracy (top) and the self-verification errors (bottom).

Since reflective SFT mixes the same CoT examples as used in non-reflective SFT, an explanation for this phenomenon is that learning to self-verify serves as a *regularizer* to the planning policy. This substantially improves the quality of hidden embeddings in transformers, which facilitates the learning of CoT examples. Binary verification is inherently a harder target to learn, which produces stronger regularizing effects than detailed verification. However, the complexity (length) of the verification should match the capacity of the model; otherwise, it could severely compromise the benefits of learning self-verification. For instance, learning binary verification and detailed verification fails to improve the 16M model and the 1M model, respectively.

**When do reflective executions improve reasoning accuracy?** Figure 7 evaluates the non-reflective, RMTP, and RTBS executions for models in solving ID-Hard problems. Apart from the accuracy, the rates of verification error (i.e., the *false positive rate* $e_+$ and *false negative rate* $e_-$ defined in Section 4) are measured using an oracle verifier. In these results, RMTP reasoning raises the performance over non-reflective reasoning except for the 1M models (which fail in ID-hard Sudoku). Smaller error rates (especially $e_-$) generally lead to higher improvements, whereas a high $e_-$ in binary verification severely compromises the performance of the 1M Mult Model. Overall, *reflection improves reasoning if the chance of rejecting correct steps ($e_-$) is sufficiently small*.

**In what task is the trace-back search helpful?** As seen in Figure 7, though RTBS shows no advantage against RMTP in Mult, it outperforms RMTP in Sudoku, especially the 4M model with detailed verification. This aligns with Theory 1 — The state of Sudoku (the $9 \times 9$ matrix) is required to comply with explicit verifiable rules, making incorrect states easily discriminated from correct states. However, errors in Mult states can only be checked by recalculating all historical steps. Therefore, we are more likely to have $f > \alpha$ in Sudoku, which grants a higher chance of solving harder problems. This suggests that *RTBS can be more helpful than RMTP if incorrect states in the task carry verifiable errors*, which validates our theoretical results.

## 5.2 Results of reinforcement learning

As introduced in Section 3.3, we further apply GRPO to fine-tune the models after SFT. Especially, GRPO based on RMTP allows solution planning and verification to be jointly optimized for self-verifying policies. The full GRPO results are presented in Appendix E.3, and the main findings are presented below. Overall, RL does enable most models to better solve ID problems, yet such improvements arise from a superficial shift in the distribution of known reasoning skills.

**How does RL improve reasoning accuracy?** Figure 8 presents the performance of 4M and 16M models in Mult after GRPO, where the differences from SFT results are visualized. GRPO effectively enhances accuracy in solving ID-Hard problems, yet the change in OOD performance is marginal. Therefore, *RL can optimize ID performance, while failing to generalize to OOD cases*.

**Does RL truly enhance verification?** From the change of verification errors in Figure 8, we find that the false negative rate $e_-$ decreases along with an increase in the false positive rate $e_+$. This suggests

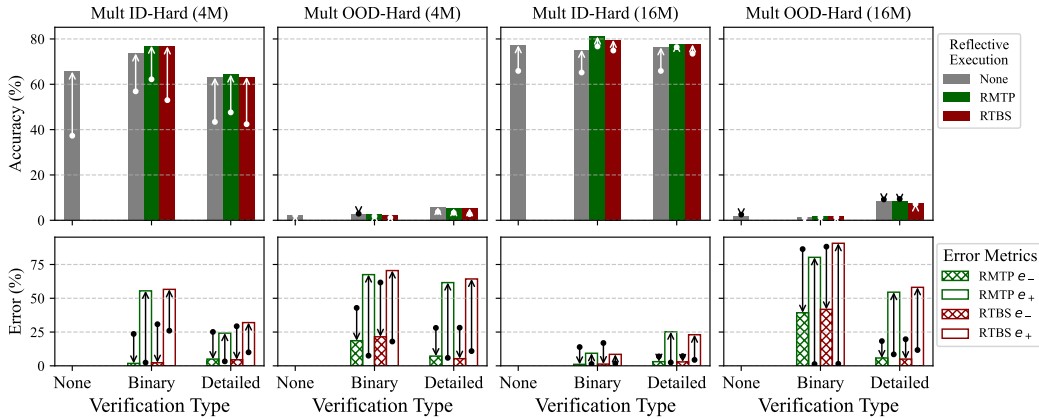

Figure 8: Performance of the 4M and 16M models in Mult after GRPO, including accuracy and the verification error rates. As an ablation, we also include non-reflective models. The vertical arrows start from the baseline accuracy after SFT, presenting the relative change caused by GRPO.

that models learn an *optimistic bias*, which avoids rejecting correct steps through a high false positive rate that bypasses verification. In other words, *instead of truly improving the verifier* (where $e_-$ and $e_+$ both decrease), *RL mainly induces an error-type trade-off*, shifting from false negatives ($e_+$) to false positives ($e_-$).

To explain this, we note that a high $e_-$ raises the computational cost (Proposition 1) and thus causes a significant performance loss under the limited budget of RL sampling, making reducing $e_-$ more rewarding than maintaining a low $e_+$. Meanwhile, shifting the error type is easy to learn, achievable by adjusting only a few parameters in the output layer of the transformer.

Inspired by DeepSeek-R1 [5], we additionally examine how RL influences the frequency of reflective behavior. To simulate the natural distribution of human reasoning, we train models to perform **optional detailed verification** by adding examples of *empty verification* (in the same amount as the full verification) into reflective SFT. This allows the policy to optionally omit self-verification, usually with a higher probability than producing full verification, since empty verification is easier to learn. Consequently, we can measure the *reflection frequency* by counting the proportion of steps that include non-empty verification. Since models can implicitly omit binary verification by producing false positive labels, we do not explicitly examine the optional binary verification.

**When does RL incentivize frequent reflection?** Figure 9 shows reflection frequency in Mult before and after GRPO, comparing exploratory (1.25) and exploitative (1) temperatures when sampling experience CoTs. With a temperature 1.25, GRPO elicits frequent reflection, especially on hard queries. However, reflection frequency remains low if using temperature 1. Additional results for other model sizes and Sudoku appear in Appendix E.3.3. In conclusion, *RL can adapt reflection frequency to align with the exploratory ability of the planning policy $\pi$, encouraging more reflection if the policy can potentially explore rewards*. This helps explain why RL promotes frequent reflection in LLMs [5], as the flexibility of language naturally fosters exploratory reasoning.

## 6 Conclusion and Discussion

In this paper, we provide a foundational analysis of self-verifying reflection in multi-step CoTs using small transformers. Through minimalistic prototypes of reflective reasoning (the RMTP and RTBS), we demonstrate that self-verification benefits both training and execution. Compared to natural-language reasoning based on LLMs, the proposed minimalistic framework performs effective reasoning and reflection using limited computational resources. We also show that RL fine-tuning can enhance the performance in solving in-distribution problems and incentivize reflective thinking for exploratory reasoners. However, the improvements from RL rely on shallow patterns and lack generalizable new skills. Overall, we suggest that self-verifying reflection is inherently beneficial for CoT reasoning, yet its synergy with RL fine-tuning is limited in superficial statistics.

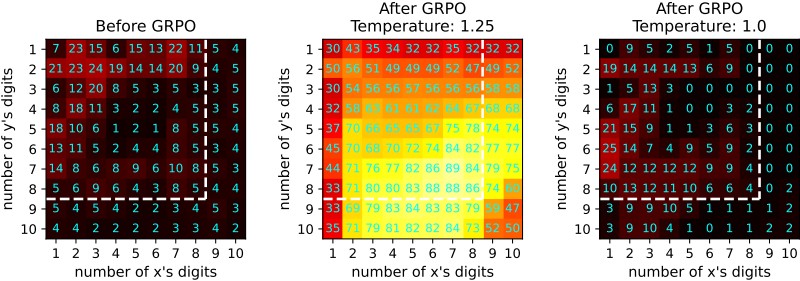

Figure 9: The hot-maps of reflection frequencies of the 4M transformer in multiplication before and after GRPO using temperatures $1$ and $1.25$, tested with RMTP execution. The $i$-th row and $j$-th column shows the frequency (%) for problems $x \times y$ where $x$ has $j$ digits and $y$ has $i$ digits.

**Limitations and future work**  Although the current training pipeline enables tiny transformers to reason properly through reflective CoTs, the *generalization ability* is still low and not improved in RL. Therefore, future work will extend reflection frameworks and explore novel training approaches. Observing the positive effect of learning self-verification, a closer connection between generative and discriminative reasoning may be the key to addressing this challenge. Additionally, how our findings *transfer from small transformers to natural-language LLMs* needs to be further examined. However, the diversity of natural language and high computational cost pose significant challenges to comprehensive evaluation, and our proposed framework does not sufficiently exploit the emergent linguistic ability of LLMs. To this end, we expect to investigate a more flexible self-verification framework with an efficient evaluator of natural-language reflection in future work.

## Acknowledgments and Disclosure of Funding

We gratefully acknowledge Dr. *Linyi Yang* for providing partial computational resources.

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

# Contents

## A   Notations

The notations used in the main paper are summarized in Table 1. Notations only appear in the appendix are not included.

Table 1: Notations in the main paper.

| Symbol | Meaning |
|---|---|
| $Q$ | The query of CoT reasoning |
| $\{R\}$ | The sequence of intermediate reasoning steps |
| $R_t$ | The $t$-th intermediate step in CoT reasoning |
| $A$ | The answer of CoT reasoning. |
| $T$ | The number of steps (including the final answer) in an CoT |
| $\pi$ | The planning policy in MTP reasoning |
| $s_t$ | The $t$-th state in CoT reasoning |
| $\mathcal{T}$ | The transition function in an MTP |
| "✓" | The special token as the positive label of verification. |
| "×" | The special token as the negative label of verification |
| $V_t$ | The verification sequence for the proposed step $R_t$. |
| $\mathcal{V}$ | The verifier such that $V_{t+1} \sim \mathcal{V}(\cdot\|S_t, R_{t+1})$ |
| $\tilde{R}_t$ | The verified reasoning step, i.e. $(R_t, V_t)$ |
| $\tilde{\mathcal{T}}$ | The reflective transition function in an RMTP |
| $\tilde{\pi}$ | The self-verifying policy, i.e. $\{\pi, \mathcal{V}\}$ |
| $m$ | The RTBS width, i.e. maximal number of attempts on each state |
| $\mu$ | The probability of proposing a correct step on positive states |
| $e_-$ | The probability of instantly rejecting a correct step on positive states |
| $e_+$ | The probability of accepting an incorrect step on positive states |
| $f$ | The probability of instantly rejecting any step on negative states |
| $\alpha$ | The shorthand of $\mu e_- + (1-\mu)(1-e_+)$ |
| $\rho(n)$ | The accuracy of non-reflective MTP reasoning |
| $\tilde{\rho}(n)$ | The accuracy of RMTP reasoning for queries with scale $n$ |
| $\tilde{\rho}_m(n)$ | The accuracy of RTBS reasoning with width $m$ for queries with scale $n$ |

## B   Details of reinforcement learning

This section introduces PPO and GRPO algorithms used in RL fine-tuning. We introduce PPO and GRPO under the context of MTP, which is described in Section 3.1. This also applies to RMTP

reasoning in Section 3.2, as RMTP is a special MTP given the self-verifying policy $\tilde{\pi}$ and the reflective transition function $\tilde{\mathcal{T}}$.

For any sequence $X$ of tokens, we additionally define the following notations: $X^{[i]}$ denotes the $i$-th token, $X^{[<i]}$ ($X^{[\leq i]}$) denotes the former $i-1$ ($i$) tokens, and $|X|$ denotes the length (i.e., the number of tokens).

Both PPO and GRPO iteratively update the reasoning policy through online experience. Let $\pi_\theta$ to denote a reasoning policy parameterized by $\theta$. On each iteration, PPO and GRPO use a similar process to update $\theta$:

1. Randomly draw queries from the task or taring set, and apply the old policy $\pi_{\theta_{old}}$ to sample experience CoTs.

2. Use reward models to assign rewards to the experience CoTs. Let ORM and PRM be the outcome reward model and process reward model, respectively. For each CoT $(Q, R_1, \ldots, R_{T-1}, A)$, we obtain outcome rewards $r_o = \mathrm{ORM}(Q, A)$ and the process rewards $r_t = \mathrm{PRM}(S_t, R_{t+1})$ for $t = 0, 1, \ldots, T-1$ (where $R_T = A$). In our case, we only use the outcome reward model and thus all process rewards are 0.

3. Then, $\theta$ is updated by maximizing an objective function based on the experience CoTs with above rewards. Especially, PPO additionally needs to update an value approximator.

## B.1 Proximal policy optimization

PPO [29] is a classic RL algorithm widely used in various applications. It includes a value model $v$ to approximate the value function, namely the expected cumulated rewards:

$$v(S_t, R_t^{[<i]}) = \mathbb{E}_\pi \left( r_o + \sum_{k=t}^{T} r_k \right) \tag{7}$$

Let $q_{t,i}(\theta) = \frac{\pi_\theta \left( R_t^{[i]} \big| S_t, R_t^{[<i]} \right)}{\pi_{\theta_{old}} \left( R_t^{[i]} \big| S_t, R_t^{[<i]} \right)}$ be the relative likelihood of the $i$-th token in the $t$-th step, and $\pi_{ref}$ be the reference model (e.g., the policy before RL-tuning). Then, the PPO algorithm maximizes

$$
\begin{aligned}
J_{PPO}(\theta) =& \mathbb{E}_{Q \sim P(Q), \{R\} \sim \pi_{\theta_{old}}} \frac{1}{\sum_{t=1}^{T} |R_t|} \sum_{t=1}^{T} \sum_{i=1}^{|R_t|} \\
& \left\{ \min \left[ q_{t,i}(\theta) \hat{A}_{t,i}, \mathrm{clip} \left( q_{t,i}(\theta), 1 - \varepsilon, 1 + \varepsilon \right) \hat{A}_{t,i} \right] - \beta \mathbb{D}_{KL} \left[ \pi_\theta \| \pi_{ref} \right] \right\}.
\end{aligned} \tag{8}
$$

Here, $\hat{A}_{t,i}$ is the advantage of the $i$-th token in step $t$, computed using the value model $v$. For example, $\hat{A}_{t,i} = v(S_t, R_t^{[<i]}, R_t^{[i]}) - v(S_t, R_t^{[<i]})$ is a simple way to estimate advantage. In practice, advantages can be estimated using the general advantage estimation (GAE) [28].

The value model $v$ is implemented using *the same architecture as the reasoner except for the output layer*, which is replaced by a linear function that outputs a scalar value. The value model is initialized using the same parameters as the reasoner, apart from the output layer. Assuming that $v$ is parameterized by $\omega$, we learn $v$ by minimizing the temporal-difference error:

$$J_v(\omega) = \mathbb{E}_{Q \sim P(Q), R \sim \pi_{\theta_{old}}} \sum_{t=1}^{T} \sum_{i=1}^{|R_t|} \left( v_\omega(S_t, R_t^{[<i]}) - v_{\omega_{old}}(S_{t+1}) \right)^2. \tag{9}$$

Although PPO proves effective in training LLMs [23], we deprecate using it in training tiny transformers due to the difficulty of learning the value function. Since the value model $v$ is also a tiny transformer, its weakness in model capacity severely compromise the precision of value approximation, leading to unreliable advantage estimation.

## B.2 Group-reward policy optimization

PPO requires learning an additional value model, which can be expensive and unstable. Alternatively, GRPO [31] directly computes the advantages using the relative rewards from a group of $G$ solutions.

For each query $Q$, it samples a group of $G$ solutions:

$$\{R_g\} = (R_{g,1}, \ldots, R_{g,T_g-1}, A_g) \sim \pi_{\theta_{old}}, \qquad \text{for } g = 1, \ldots, G. \tag{10}$$

In this group, each solution $\{R_g\}$ contains $T_g$ steps, where the answer $A_g$ is considered as the final step $R_{g,T_g}$. Using the reward models, we obtain process rewards $\boldsymbol{r}_p := \{(r_{g,1}, \ldots, r_{g,T_g})\}_{g=1}^{G}$ and outcome rewards $\boldsymbol{r}_o := \{r_{g,o}\}_{g=1}^{G}$. Then, GPRO computes the normalized rewards, given by:

$$\tilde{r}_{g,t} = \frac{r_{g,t} - \text{mean } \boldsymbol{r}_p}{\text{std } \boldsymbol{r}_p}, \ \tilde{r}_{g,o} = \frac{r_{g,o} - \text{mean } \boldsymbol{r}_o}{\text{std } \boldsymbol{r}_o} \tag{11}$$

Afterwards, the advantage of step $t$ in the $g$-th solution of the group is $\hat{A}_{g,t} = \tilde{r}_{g,o} + \sum_{k=t}^{T_g} \tilde{r}_{g,t'}$. Let $q_{g,t,i}(\theta) = \frac{\pi_\theta\left(R_{g,t}^{[i]} \middle| S_{g,t}, R_{g,t}^{[<i]}\right)}{\pi_{\theta_{old}}\left(R_{g,t}^{[i]} \middle| S_{g,t}, R_{g,t}^{[<i]}\right)}$ be the relative likelihood of the $i$-th token in the $t$-th step from the $g$-th solution. Then, the GRPO objective is to maximize the following:

$$J_{GRPO}(\theta) = \mathbb{E}_{Q \sim P(Q), \{R_g\} \sim \pi_{old}} \frac{1}{G} \sum_{g=1}^{G} \frac{1}{\sum_{t=1}^{T_g} |\tau^{(t)}|} \sum_{t=1}^{T_g} \sum_{i=1}^{|R_{g,t}|}$$

$$\left\{ \min\left[ q_{g,t,i}(\theta)\hat{A}_{g,t}, \text{clip}\left(q_{g,t,i}(\theta), 1-\varepsilon, 1+\varepsilon\right)\hat{A}_{g,t} \right] - \beta \mathbb{D}_{KL}\left[\pi_\theta \| \pi_{ref}\right] \right\} \tag{12}$$

### B.3 Technical Implementation

We made two technical modifications that make RL more suitable in our case, described in the following.

First, in RMTP, we *mask off the advantage of rejected steps*, while the advantage of self-verification labels is reserved. This prevents the algorithm from increasing the likelihood of rejected steps, allowing the planning policy $\pi$ to be properly optimized. In practice, we find this modification facilitates the training of models that perform mandatory detailed verification. Otherwise, RL could make the reasoner excessively rely on reflection, leading to CoTs that are unnecessarily long.

Second, we employ an *early-truncating strategy* when sampling trajectories in training. If the model has already made a clear error at some step (detected using an oracle process reward model), we truncate the trajectory as it is impossible to find a correct answer. This avoids unnecessarily punishing later steps due to previous deviations, as some later steps may be locally correct in their own context. Empirically, we find this modification reduces the training time required to reach the same performance, while the difference in final performance is marginal.

## C Theory

### C.1 A general formulation of reasoning performance

Let $\mathcal{S}$ denote the state space and $\mathcal{A}$ denote the answer space. We use $\mathcal{A}_Q \subseteq \mathcal{A}$ to denote the set of correct answers for some input query $Q$. Given any thought state $S$, the accuracy, namely the probability of finding a correct answer, is denoted as

$$\rho_Q(S) = p_{(R_{t+1}, R_{t+2}, \ldots, A) \sim \pi}(A \in \mathcal{A}_Q \mid S_t = S) \tag{13}$$

#### C.1.1 Bellman equations in RMTP

By considering the reasoning correctness as the binary outcome reward, we may use Bellman equations [33] to provide a general formulation of the reasoning performance for arbitrary MTPs (RMTP). For simplicity, we use $S$, $R$, and $S'$ to respectively denote the state, step, and next state in a transition.

Initially, in the absence of a trace-back mechanism, the accuracy $\rho_Q(s)$ can be interpreted as the value function when considering the MTP as a goal-directed decision process. For simplicity, we denote the transition probability drawn from the reasoning dynamics $\mathcal{T}$ as $p(S' \mid S, R)$. In non-reflective reasoning, the state transition probability $p(S' \mid S)$ can be expressed as:

$$p(S'|S) = \sum_R p(S'|S, R)\pi(R|S) \tag{14}$$

When using RMTP execution, assuming that $\xi(S, R) := p_{V \sim \mathcal{V}}(\text{``}\times\text{''} \in V \mid S, R)$ represents the probability of rejecting the step $R$, we have:

$$p(S'|S) = \begin{cases} \sum_R \pi(R|S)(1 - \xi(S, R))p(S'|S, R), & \text{if } S' \neq S \\ \sum_R \pi(R|S)\left((1 - \xi(S, R))p(S'|S, R) + \xi(S, R)\right), & \text{if } S' = S \end{cases} \quad (15)$$

Consequently, the Bellman equation follows:

$$\rho_Q(S) = \begin{cases} 1, & \text{if } S \in \mathcal{A}_Q \\ 0, & \text{if } S \in \mathcal{A} \setminus \mathcal{A}_Q \\ \sum_{S'} \rho_Q(S')p(S' \mid S), & \text{if } s \in \mathcal{S} \setminus \mathcal{A} \end{cases} \quad (16)$$

### C.1.2 Bellman equations in RTBS

Let $m$ denote the number of attempts at each state, and let $\phi(S)$ represent the failure probability (i.e., the probability of tracing back after $m$ rejected steps) at state $S$. The probability of needing to retry a proposed step due to instant rejection or recursive rejection is given by:

$$\epsilon(S) = \sum_R \pi(R|S) \left( \xi(S, R) + \sum_{S'}(1 - \xi(S, R))p(S'|S, R)\phi(S') \right) \quad (17)$$

The failure probability is then given by $\phi(S) = \epsilon^m(S)$. When there are $k$ attempts remaining at the current state $S$, we denote the accuracy as $\rho_x(S, k)$, given by:

$$\rho_Q(S, k) = \begin{cases} \epsilon(S)\rho_x(S, k - 1) + \sum_R \pi(R|S)(1 - \xi(S, R)) \sum_{S'} p(S'|S, R)\rho_Q(S'), & k > 0 \\ 0, & k = 0 \end{cases} \quad (18)$$

It follows that $\rho_x(S) = \rho_x(S, m)$. This leads to a recursive formulation that ultimately results in the following equations for each $s \in \mathcal{S}$:

$$\epsilon(S) = \sum_R \pi(R|S) \left( \xi(S, R) + \sum_{S'}(1 - \xi(S, R))p(S'|S, R)\epsilon^m(S') \right), \quad (19)$$

$$\rho_x(S) = \frac{1 - \epsilon^m(S)}{1 - \epsilon(S)} \sum_{S'} \rho_Q(S')\pi(R|S)(1 - \xi(S, R))p(S'|S, R). \quad (20)$$

### C.2 Accuracy derivation in the simplified reasoning task

In the following, we derive the accuracy of reflective reasoning with and without the trace-back search, given the simplified reasoning task in Section 4. For each proposed step on a correct state, we define several probabilities to simplify notations: $\alpha := \mu e_- + (1 - \mu)(1 - e_+)$ is the probability of being **instantly rejected**; $\beta = \mu(1 - e_-)$ is the probability of being **correct and accepted**; $\gamma = (1 - \mu)e_+$ is the probability of being **incorrect but accepted**. Note that $\beta$ here no longer refers to the KL-divergence factor in Appendix B.

**Proposition 2.** *The RTMP accuracy $\tilde{\rho}(n)$ for problems with a scale of $n$ is*

$$\tilde{\rho}(n) = \left( \frac{\beta}{1 - \alpha} \right)^n \quad (21)$$

*Let $m$ be the width of RTBS. Let $\delta_m(n)$ and $\epsilon_m(n)$ be the probability of a proposed step being rejected (either instantly or recursively) on a correct state and incorrect state of scale $n$, respectively. We have $\sigma_m(0) = \epsilon_m(0) = 0$ and the following recursive equations for $n > 0$:*

$$\delta_m(n) = \alpha + \beta(\delta_m(n - 1))^m + \gamma(\epsilon_m(n - 1))^m \quad (22)$$

$$\epsilon_m(n) = f + (1 - f)(\epsilon_m(n - 1))^m \quad (23)$$

*Then, the RTBS accuracy $\tilde{\rho}_m(n)$ for problems with a scale of $n$ is given by*

$$\tilde{\rho}_m(n) = \prod_{t=1}^n \sigma_m(t), \qquad \text{where } \sigma_m(t) = \beta \sum_{i=0}^{m-1} (\delta_m(t))^i = \frac{1 - (\delta_m(t))^m}{1 - \delta_m(t)}\beta. \quad (24)$$

*In addition, $\delta_m(n)$, $\epsilon_m(n)$ and $\sigma_m(n)$ all motonously increase and converge in relation to $n$.*

*Proof.* We first consider reasoning through RTBS. Let $\phi_m(n)$ and $\psi_m(n)$ denote the probabilities of failure (reaching the maximum number of attempts) in correct and incorrect states, respectively. Let $\tilde{\rho}_{i|m}(n)$ indicate the accuracy after the $i$ attempts at the current sub-problem of scale $n$. Therefore, we have $\tilde{\rho}_m(n) = \tilde{\rho}_{0|m}(n)$ and $\tilde{\rho}_{m|m}(n) = 0$.

At a correct state, we have the following possible cases:

- A correct step is proposed and instantly accepted with probability $\beta = \mu(1 - e_-)$. In this case, the next state has a scale of $n - 1$, which is correctly solved with probability $\rho_{0|m}(n - 1)$ and fails (i.e., is recursively rejected) with probability $\phi_m(n - 1)$.

- A correct step is proposed and instantly rejected with probability $\mu e_-$.

- An incorrect step is proposed and instantly accepted with probability $\gamma = (1 - \mu)e_+$. In this scenario, the next state has a scale of $n - 1$, which fails with probability $\psi_m(n - 1)$.

- An incorrect step is proposed and instantly rejected with probability $\beta = (1 - \mu)(1 - e_+)$.

Thus, we have a probability of $\alpha = \mu e_- + (1 - \mu)(1 - e_+)$ to instantly reject the step, and a probability of $\beta \phi_m(n - 1) + \gamma \psi_m(n - 1)$ to recursively reject the step. Therefore, the overall probability of *rejecting an attempt on correct states* is:

$$\delta_m(n) = \alpha + \beta \phi_m(n - 1) + \gamma \psi_m(n - 1). \tag{25}$$

Since failure occurs after $m$ rejections, we have:

$$\phi_m(n) = (\delta_m(n))^m \tag{26}$$

At an incorrect state, we have a probability $f$ to instantly reject a step. Otherwise, we accept the step, and the next state fails with probability $\psi_m(n - 1)$. Therefore, the overall probability of rejecting an attempt for incorrect states is:

$$\epsilon_m(n) = f + (1 - f)\psi_m(n - 1). \tag{27}$$

Similarly, we obtain:

$$\psi_m(n) = (\epsilon_m(n))^m \tag{28}$$

By substituting Equations (26) and (28) into Equations (25) and (27), we obtain Equations (22) and (23). If an attempt is rejected (either instantly or recursively), we initiate another attempt which solves the problem with a probability of $\rho_{i+1|m}(n)$. Therefore, we have the recursive form of the accuracy, given by:

$$\tilde{\rho}_{i|m}(n) = \beta \tilde{\rho}_{0|m}(n - 1) + \delta(n, m)\tilde{\rho}_{i+1|m}(n) \tag{29}$$

Thus, we can expand $\tilde{\rho}_m(n)$ as:

$$\tilde{\rho}_m(n) = \tilde{\rho}_{0|m}(n)$$
$$= \beta \tilde{\rho}_m(n - 1) + \delta_m(n)\tilde{\rho}_{1|m}(n)$$
$$\cdots$$
$$= (\beta + \delta_m(n)\beta + \delta_m^2(n)\beta + \cdots + \delta_m^m(n)\beta)\tilde{\rho}_m(n - 1) \tag{30}$$
$$= \sigma_m(n)\tilde{\rho}_m(n - 1) \tag{31}$$

Note that $n = 0$ indicates that the state is exactly the outcome, which means $\tilde{\rho}_m(0) = 1$. Then, Equation (24) is evident given the recursive form in Equation (31).

For reflective reasoning **without trace-back**, we can simply replace $\delta_m(n)$ with $\alpha$ in $\sigma_m(n)$, as only instant rejections are allowed. We then set $m \to \infty$, leading to Equation (21).

**Monotonicity** We first prove the monotonic increase of $\epsilon_m(n)$. Equation (23) gives $\epsilon_m(n) = f + (\epsilon_m(n-1))^m$ and $\epsilon_m(n+1) = f + (\epsilon_m(n))^m$ for each $n > 1$. Therefore, if $\epsilon_m(n) \geq \epsilon_m(n-1)$, we have:

$$\epsilon_m(n + 1) = f + (\epsilon_m(n))^m \geq f + (\epsilon_m(n - 1))^m = \epsilon_m(n). \tag{32}$$

Additionally, it is clear that $\epsilon_m(1) = f \geq 0 = \epsilon_m(0)$. Using mathematical induction, we conclude that $\epsilon_m(n + 1) > \epsilon_m(n)$ for all $n \geq 0$. The monotonicity of $\delta_m(n)$ can be proven similarly, and the monotonicity of $\sigma_m(n)$ is evident from that of $\delta_m(n)$. Since $\delta_m(n)$ and $\epsilon_m(n)$ are probabilities, they are bounded in $[0, 1]$ and thus converge monotonically. $\square$

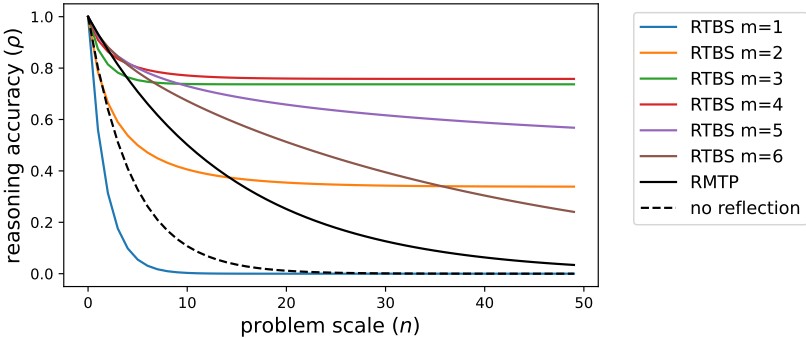

Figure 10: The accuracy curves of non-reflective MTP $\rho(n)$, RMTP $\tilde{\rho}(n)$, and RTBS $\tilde{\rho}_m(n)$, using $\mu = 0.8$, $e_- = 0.3$, $e_+ = 0.2$, and $f = 0.8$.

**Illustration of accuracy curves** Using the recursive formulae in Proposition 2, we are able to implement a program to compute the reasoning accuracy in the simplified reasoning problem in Section 4 and thereby visualize the accuracy curves of various reasoning algorithms. For example, Figure 10 presents the reasoning curves given $\mu = 0.8$, $e_- = 0.3$, $e_+ = 0.2$, and $f = 0.8$, which lead to $\alpha = 0.4 < f$. For this example, we may observe the following patterns: 1) An overly small width $m$ in RTBS leads to poor performance; and 2) by choosing $m$ properly, $\tilde{\rho}_m(n)$ remains stable when $n \to \infty$. These observations are formally described and proved in Appendix C.3.

Furthermore, in Figure 10 we see that a small $m$ stabilizes the drop of $\tilde{\rho}_m(n)$ when $n$ is large, yet it also makes $\tilde{\rho}_m(n)$ drop sharply in the area where $n$ is small. This indicates the potential of using an *adaptive width* in RTBS, where $m$ is set small when the current subproblem (state) requires a large number $n$ of steps to solve, and $m$ increases when $n$ is reduced by previous reasoning steps. Since this paper currently focuses on the minimalistic reflection framework, we expect to explore such an extension in future work.

### C.3 Derivation of Theorem 1

Theorem 1 is obtained by merging the following Proposition 3 and Proposition 4, which also provide supplementary details on the non-trivial assumptions of factors $\mu$, $e_-$, and $e_+$. Additionally, Proposition 4 also shows that there exists an ideal range of RTBS width $m$ such that stabilizes the drop of $\tilde{\rho}_m(n)$ when $n \to \infty$.

**Proposition 3** (RMTP Validity conditions). *For all $n \geq 0$, we have $\tilde{\rho}(n) \geq \rho(n) \iff e_- + e_+ \leq 1$. Additionally, if $\mu > 0$ and $e_- < 1$, then for all $n \geq 1$ we have that $\tilde{\rho}(n) = \rho(n) \iff e_- + e_+ = 1$ and $\tilde{\rho}(n)$ decreases strictly with either $e_-$ or $e_+$.*

**Proposition 4** (RTBS Validity Condition). *Assuming $0 < \mu < 1$, $e_- < 1$, and $e_+ > 0$, then*

$$\lim_{n \to \infty} \frac{\tilde{\rho}_m(n)}{\tilde{\rho}(n)} > 1 \iff \left( m > \frac{1}{1-\alpha} \text{ and } f > \alpha \right). \tag{33}$$

*Furthermore, we have*

- $\lim_{n \to \infty} \frac{\tilde{\rho}_m(n)}{\tilde{\rho}_m(n-1)} = 1$ *if* $m \in [\frac{1}{\mu(1-e_-)}, \frac{1}{1-f}]$.
- $\tilde{\rho}_m(n)$ *increases strictly with $f$ for all $n \geq 2$.*

The proof of propositions 3 and 4 is given in Appendix C.3.1 and Appendix C.3.2, which are based on the previous derivation in Appendix C.2.

### C.3.1 Proof of Proposition 3

In any case, we have $\tilde{\rho}(0) = \rho(0) = 1$.

If $\mu = 0$ or $e_- = 1$, we clearly have $\tilde{\rho}(n) = \rho(n) = 0$ for $n \geq 1$.

If $0 > \mu$ and $e_- < 1$, we can transform $\tilde{\rho}(n)$ (given in Proposition 2) as:

$$\tilde{\rho}(n) = \left( \frac{1}{1 + \frac{e_+}{1-e_-}(\mu^{-1}-1)} \right)^n = \left( \frac{\mu(1-e_-)}{\mu(1-e_+-e_-)+e_+} \right)^n. \tag{34}$$

This shows that $\rho(n)$ strictly decreases with both $e_+$ and $e_-$, and $\tilde{\rho}(n) = \mu^n \iff e_+ + e_- = 1$. Therefore, we also have $\tilde{\rho}(n) > \mu^n \iff e_+ + e_- < 1$.

The Proposition is proved by combining all the above cases.

### C.3.2 Proof of Proposition 4

The assumptions $0 < \mu < 1$, $e_- < 1$, and $e_+ > 0$ ensure that $\beta > 0$ and $\gamma > 0$. Proposition 2 suggests the monotonous convergence of $\delta_m(n)$, $\epsilon_m(n)$, and $\sigma_m(n)$. For simplicity, we denote $\delta := \lim_{n \to \infty} \delta_m(n)$, $\epsilon := \lim_{n \to \infty} \epsilon_m(n)$, and $\sigma := \lim_{n \to \infty} \sigma_m(n)$. From Equations (22) and (23), we have:

$$\delta = \alpha + \beta\delta^m + \gamma\epsilon^m \tag{35}$$
$$\epsilon = f + (1-f)\epsilon^m \tag{36}$$

Note that $\epsilon = \delta = 1$ gives the trivial solution of the above equations. However, there may exist another solution (if any) such that $\delta < 1$ or $\epsilon < 1$ under certain circumstances. Since $\epsilon_m(0) = 0$ and $\delta_m(0) = 0$, the limits $\epsilon$ and $\delta$ take the smaller solution within $[0, 1]$. In the following, we first discuss when another non-trivial solution exists.

**Lemma 1.** For any $m \geq 1$, if $0 \leq p < \frac{m-1}{m}$, then $x = p + (1-p)x^m$ has a unique solution $x_* \in [p, 1)$, which strictly increases with $p$. Otherwise, if $\frac{m-1}{m} \leq p \leq 1$, the only solution in $[0, 1]$ is $x_* = 1$.

*Proof.* Define $F(x) := p + (1-p)x^m - x$. We find:

$$F'(x) = m(1-p)x^{m-1} - 1.$$

It is observed that $F'(x)$ increases monotonically with $x$. Additionally, we have $F'(0) = -1 < 0$, $F'(1) = m(1-p) - 1$, and $F(1) = 0$. We only consider the scenario where $p > 0$, since $x = 0 \in [0, 1)$ is obviously the unique solution.

If $0 \leq p < \frac{m-1}{m}$, we have $1 - p > \frac{1}{m}$. This implies $F'(1) > 0$. Combining $F'(0) < 0$, there exists $\xi \in (0, 1)$ such that $F'(\xi) = 0$. As a result, $F(x)$ strictly decreases in $[0, \xi]$ and increases in $[\xi, 1)$. Therefore, we have $F(\xi) < F(1) = 0$. Since $F(p) = (1-p)p^m > 0$, we know that there exists a unique $x_* \in [p, \xi]$ such that $F(x_*) = 0$.

If $\frac{m-1}{m} \leq p \leq 1$, we have $1 - p \leq \frac{1}{m}$ and $F'(1) \leq 0$. In this case, $F'(x) < 0$ in $[0, 1)$ due to the monotonicity of $F'(x)$. Thus, $F(x) > F(1) = 0$ for all $x \in [0, 1)$. Therefore, $x_* = 1$ is the only solution within $[0, 1]$.

Now, we prove the monotonic increase of $x_*$ when $0 \leq p < \frac{m-1}{m}$. We have:

$$\frac{\mathrm{d}x_*}{\mathrm{d}p} = 1 + m(1-p)x_*^{m-1}\frac{\mathrm{d}x_*}{\mathrm{d}p} - x_*^m \tag{37}$$

$$\frac{\mathrm{d}x_*}{\mathrm{d}p} = \frac{1 - x_*^m}{1 - m(1-p)x_*^{m-1}} = \frac{1 - x_*^m}{-F'(x_*)} \tag{38}$$

The previous discussion shows that with $x_* < [p, \xi]$ for some $\xi$ such that $F'(\xi) = 0$. Given that $F'(x)$ increases monotonically, we have $F'(x_*) < 0$ and thus $\frac{\mathrm{d}x_*}{\mathrm{d}p} > 0$. □

**Lemma 2.** Assume $p \geq 0$, $q > 0$, and $p + q \leq 1$. Then, the equation $x = p + qx^m$ has a unique solution $x_* \in [0, 1)$, which increases monotonically with $p \in [0, 1 - q]$.

*Proof.* Define $F(x) := p + qx^m - x$. Since $F(0) \geq 0$ and $F(1) < 0$, there exists a solution $x_* \in [0, 1)$. Since $F$ is convex, we know there is at most one other solution. Clearly, the other solution appears in $(1, +\infty)$, since $F(+\infty) > 0$. Therefore, $F(x) = 0$ must have a unique solution $x_*$ in $[0, 1)$. Additionally, $x_*$ must appear to the left of the minimum of $F$, which yields $F'(x_*) < 0$.

Using the Implicit Function Theorem, we write:

$$\frac{\mathrm{d}x_*}{\mathrm{d}p} = \frac{1}{1 - mqx_*^{m-1}} = -\frac{1}{F'(x_*)} > 0 \tag{39}$$

Thus, we conclude that $x_*$ increases monotonically with $p$. $\qquad\square$

Applying Lemmas 1 and 2 to Equation (36), we find that $\epsilon = 1$ if and only if $f \geq \frac{m-1}{m}$; otherwise, $\epsilon$ strictly increases with $p$. Therefore, $f < \frac{m-1}{m}$ indicates that $\epsilon < 1$, leading to $(\alpha + \gamma\epsilon) + \gamma < 1$. Using Lemma 2 again in Equation (35), we have $f < \frac{m-1}{m} \implies \delta < 1$. Conversely, $f \geq \frac{m-1}{m}$ yields $\epsilon = 1$. and thus $f, \alpha + \gamma \geq \frac{m-1}{m} \implies \delta = 1$.

First, we consider the special case where $\delta = 1$, which occurs if both $f, \alpha + \gamma \geq \frac{m-1}{m}$, namely $m \leq \min\{\frac{1}{1-f}, \frac{1}{\beta}\}$. In this case, we write $\sigma = (1 + \delta + \cdots + \delta^{m-1})\beta = m\beta$. Therefore, we have:

$$\lim_{n\to\infty} \frac{\tilde{\rho}(n)}{\rho(n)} > 1 \iff \sigma > \frac{\beta}{1-\alpha}$$
$$\iff m\beta > \frac{\beta}{1-\alpha}$$
$$\iff m > \frac{1}{1-\alpha}$$

This leads to the validity condition that $\frac{1}{1-\alpha} < m \leq \min\{\frac{1}{1-f}, \frac{1}{\beta}\}$.

Next, we consider the case where $\delta < 1$, which occurs when $f < \frac{m-1}{m}$ or $\alpha + \gamma < \frac{m-1}{m}$. This leads to $\beta \geq \frac{1}{m} > 0$, and we can write:

$$\delta^m = \frac{1}{\beta}\left(\delta - \alpha - \gamma\epsilon^m\right), \tag{40}$$

$$\sigma = \frac{1 - \delta^m}{1 - \delta} = \frac{(1 - \delta^m)\beta}{(1-\alpha) - (\beta\delta^m + \gamma\epsilon^m)}. \tag{41}$$

Then, we can derive:

$$\lim_{n\to\infty} \frac{\tilde{\rho}(n)}{\rho(n)} < 1 \iff \sigma > \frac{\beta}{1-\alpha}$$
$$\iff \frac{1 - \delta^m}{(1-\alpha) - (\beta\delta^m + \gamma\epsilon^m)} > \frac{1}{1-\alpha}$$
$$\iff (1 - \delta^m)(1 - \alpha) > (1-\alpha) - (\beta\delta^m + \gamma\epsilon^m)$$
$$\iff \gamma\epsilon^m > \delta^m(1 - \alpha - \beta)$$
$$\iff \gamma\epsilon^m > \gamma\delta^m$$
$$\iff \epsilon > \delta$$

Since we have assumed $\delta < 1$, we have $\epsilon = 1 > \delta$ if $f \geq \frac{m-1}{m}$; otherwise if $f = \alpha < \frac{m-1}{m}$, then $\epsilon = \delta$ leads to $\delta = \epsilon$ being a solution of Equation (35). Additionally, from Lemmas 1 and 2, we know that a higher $\alpha$ would increase $(\alpha + \gamma\epsilon)$, which eventually raises $\delta$ above $\epsilon$; conversely, a lower $\alpha$ causes $\delta$ to drop below $\epsilon$. To summarize, we have the following conditions when $\delta < 1$:

$$1 \geq \epsilon > \delta \iff \left(\alpha + \gamma < \frac{m-1}{m} \leq f\right) \text{ or } \left(\alpha < f < \frac{m-1}{m}\right) \tag{42}$$

$$\iff \left(\frac{1}{\beta} < m \leq \frac{1}{1-f}\right) \text{ or } \left(\alpha < f \text{ and } m > \frac{1}{1-f}\right) \tag{43}$$

Combining the conditions when $\delta = 1$, we have:

$$\lim_{n\to\infty} \frac{\tilde{\rho}(n)}{\rho(n)} > 1$$

$$\iff \left(\frac{1}{1-\alpha} < m \le \min\left\{\frac{1}{1-f}, \frac{1}{\beta}\right\}\right) \text{ or } \left(\frac{1}{\beta} < m \le \frac{1}{1-f}\right) \text{ or } \left(\alpha < f \text{ and } m > \frac{1}{1-f}\right) \tag{44}$$

$$\iff m > \frac{1}{1-\alpha} \text{ and } f > \alpha \tag{45}$$

Thus far, we have obtained Equation (33). Now, we start proving the two additional statements in Proposition 4.

**First, we prove that $\frac{1}{\beta} \le m \le \frac{1}{1-f}$ ensures that $\sigma = 1$.** First, if $\delta = 1$, we have $m \le \frac{1}{\beta} \le \frac{1}{1-f}$. In this case, $\sigma = m\beta$, and thus $\sigma = 1$ when $m = \frac{1}{\beta}$. Alternatively, if $\delta < 1$, we have $m > \min\{\frac{1}{\beta}, \frac{1}{1-f}\}$. We can express that:

$$\sigma = \frac{1-\delta^m}{1-\delta}\beta = \frac{\beta - (\delta - \alpha - \gamma\epsilon^m)}{1-\delta} = 1 - \gamma\frac{1-\epsilon}{(1-\delta)(1-f)} \tag{46}$$

Using Lemma 2, we know that $\delta$ increases with $\alpha + \gamma\epsilon$, which increases with $\epsilon$. Therefore, we have $\frac{d\delta}{d\epsilon} > 0$. Then, we obtain

$$\mathrm{d}\sigma/\mathrm{d}\epsilon = \sum_{i=1}^{m} i\delta^{m-1}\beta\frac{\mathrm{d}\delta}{\mathrm{d}\epsilon} > 0, \tag{47}$$

$$\sigma = \frac{1-\delta^m}{1-\delta}\beta = \frac{\beta - (\delta - \alpha - \gamma\epsilon^m)}{1-\delta} = \frac{\alpha + \beta + \gamma - (1-\epsilon^m)\gamma - \delta}{1-\delta}$$

$$= \frac{1 - \delta - \gamma(1 - \frac{\epsilon - f}{1-f})}{1-\delta} \tag{48}$$

Therefore, $\sigma$ increases with $\epsilon$ and reaches its maximum value of $1$ when $\epsilon = 1$. As a result, we conclude that $\frac{1}{\beta} \le m \le \frac{1}{1-f}$ ensures that $\sigma = 1$. Combining $\beta = \mu(1 - e_-)$ and $\sigma = \lim_{n\to\infty}\sigma_m(n) = \lim_{n\to\infty}\frac{\tilde{\rho}_m(n)}{\tilde{\rho}_m(n-1)}$, we have proved that $\lim_{n\to\infty}\frac{\tilde{\rho}_m(n)}{\tilde{\rho}_m(n-1)} = 1$ if $m \in [\frac{1}{\mu(1-e_-)}, \frac{1}{1-f}]$.

**Next, we prove the monotonicity of $\tilde{\rho}_m(n)$ with respect to $f$.** To prove this, we first prove the monotonicity of $\epsilon_m(t)$ for all $t$ with respect to $f$.

**Lemma 3.** For $n > 0$, $\epsilon_m(t)$ as defined in Equation 23 increases strictly with $f$.

*Proof.* We regard

$$\epsilon_m(0; f) \equiv 0, \qquad \epsilon_m(t; f) = f + (1-f)\big[\epsilon_m(t-1; f)\big]^m \tag{49}$$

as a function of $f$ on $[0, 1]$. When $t = 1$ we have

$$\epsilon_m(1; f) = f + (1-f)\big[\epsilon_m(0; f)\big]^m = f, \tag{50}$$

so $\frac{\partial\epsilon_m(1;f)}{\partial f} = 1 > 0$. Thus $\epsilon_m(1; f)$ is strictly increasing with $f$. Further, assume for some $k \ge 1$ that

$$0 \le \epsilon_m(k; f) < 1 \quad \text{and} \quad \frac{\partial\epsilon_m(k; f)}{\partial f} > 0 \quad \forall f \in [0, 1]. \tag{51}$$

Differentiate the recursion for $t = k + 1$:

$$\epsilon_m(k+1; f) = f + (1-f)\big[\epsilon_m(k; f)\big]^m, \tag{52}$$

$$\frac{\partial\epsilon_m(k+1; f)}{\partial f} = 1 - \big[\epsilon_m(k; f)\big]^m + (1-f)m\big[\epsilon_m(k; f)\big]^{m-1}\frac{\partial\epsilon_m(k; f)}{\partial f}. \tag{53}$$

By the inductive hypothesis, $\epsilon_m(k; f) < 1$ implies $\left[\epsilon_m(k; f)\right]^m < 1$. Therefore, we have

$$1 - \left[\epsilon_m(k; f)\right]^m > 0. \tag{54}$$

Since $1 - f \geq 0$, $m \geq 1$, $\left[\epsilon_m(k; f)\right]^{m-1} \geq 0$, and $\frac{\partial \epsilon_m(k; f)}{\partial f} > 0$, the second term is also positive. Hence

$$\frac{\partial \epsilon_m(k + 1; f)}{\partial f} > 0, \tag{55}$$

showing that $\epsilon_m(k + 1; f)$ is strictly increasing. This completes the induction. $\square$

Given Lemma 3, we can also prove the monotonicity of $\delta_m(t)$ using mathematical induction: It is easy to write that $\delta_m(2) = \alpha + \beta\alpha^m + \gamma f^m$, showing that $\delta_m(2)$ increases strictly with $f$. Then, for $t > 2$, assuming that $\delta_m(t - 1)$ increases strictly with $f$ and using Given Lemma 3, we know that $\delta_m(t)$ increases strictly with $f$ from Equation (22).

Therefore, we have $\delta_m(1) = \alpha$ and that $\delta_m(t)$ strictly increases with $f$ for $n \geq 2$. According to Equation (24), from the above monotonicity of $\delta_m(t)$, it is obvious that $\sigma_m(t)$ increases with respect to $f$ for all $t$. This gives the corollary that $\tilde{\rho}_m(n)$ increases with $f$ for all $n$.

### C.4 Derivation of RMTP reasoning cost

In this section, we derive how many steps it costs to find a correct solution in RMTP and thereby prove Proposition 1.

*Proposition 1.* The probability of accepting the correct step after the $i$-th attempt is given by $\alpha^{i-1}\beta$. Assuming a maximum number of attempts $m$, the number of attempts consumed at each step satisfies:

$$\Pr(i \text{ attempts} \mid \text{correct}) = \frac{\alpha^{i-1}\beta}{\beta + \alpha\beta + \cdots + \alpha^{m-1}\beta} = \frac{(1 - \alpha)\alpha^{i-1}}{1 - \alpha^m} \tag{56}$$

Therefore, the average number of attempts required for a correct reasoning step is given by

$$A_m = \sum_{i=1}^{m} i \cdot \frac{(1 - \alpha)\alpha^i}{1 - \alpha^m} = \frac{1 - \alpha}{1 - \alpha^m} \sum_{i=1}^{m} i\alpha^{i-1} \tag{57}$$

To simplify the summation expression $\sum_{i=1}^{m} i\alpha^{i-1}$ (where $0 < \alpha < 1$), we can use the method of telescoping series. Let $S = \sum_{i=1}^{m} i\alpha^{i-1}$. We calculate $\alpha S$:

$$\alpha S = \sum_{i=1}^{m} i\alpha^i \tag{58}$$

Thus,

$$S - \alpha S = \sum_{i=1}^{m} i\alpha^{i-1} - \sum_{i=1}^{m} i\alpha^i \tag{59}$$

This gives us

$$(1 - \alpha)S = 1 + \alpha + \alpha^2 + \cdots + \alpha^{m-1} - m\alpha^m = \frac{1 - \alpha^m}{1 - \alpha} - m\alpha^m \tag{60}$$

Rearranging, we have

$$\sum_{i=1}^{m} i\alpha^{i-1} = \frac{1 - \alpha^m - (1 - \alpha)m\alpha^m}{(1 - \alpha)^2} \tag{61}$$

Thus, the average number of attempts can be further expressed as:

$$A_m = \frac{1 - \alpha}{1 - \alpha^m} \cdot \frac{1 - \alpha^m - (1 - \alpha)m\alpha^m}{(1 - \alpha)^2} \tag{62}$$

$$= \frac{1 - \alpha^m - (1 - \alpha)m\alpha^m}{(1 - \alpha)(1 - \alpha^m)} \tag{63}$$

$$= \frac{1}{1 - \alpha} - m\frac{\alpha^m}{1 - \alpha^m} \tag{64}$$

$$\tag{65}$$

Considering the limit as $m \to \infty$, it can be shown using limit properties that $\lim_{m\to\infty} m \frac{\alpha^m}{1-\alpha^m} = 0$. If the correct solution is obtained (i.e., correct steps are accepted at each step), the average number of steps taken is given by

$$\bar{T} = nA_\infty = \frac{n}{1-\alpha} = \frac{n}{1 - \mu e_- - (1-\mu)(1-e_+)} = \frac{n}{(1-\mu)e_+ + \mu(1-e_-)} \tag{66}$$

$\square$

### C.5 Considering posterior risks of rejected attempts

Our previous analysis is based on a coarse binary partition of states (correct and incorrect) for each scale, which enhances clarity yet does not apply to real-world complexity. Therefore, we can introduce stronger constraints by taking into account the posterior distribution of states in $\mathcal{S}_n^+$ after multiple attempts. For example, if the state has produced several incorrect attempts on state $S$ (or rejected several correct attempts), it is more likely that the current state has not been well generalized by the model. Consequently, the chances of making subsequent errors increase. In this case, $\mu$ is likely to decrease with the number of attempts, while $e_-$ increases with the number of attempts. Thus, the probability of accepting the correct action will decrease as the number of attempts increases.

Therefore, we consider the scenario where the probabilities of error increase while the correctness rate $\mu$ drops after each attempt. We define $e_{i+}, e_{i-}, \mu_i$, etc., to represent the probabilities related to the $i$-th attempt, corresponding to the calculations of $\alpha_i, \beta_i, \gamma_i$, etc. We have that $\beta_i = \mu_i(1 - e_{i-})$ is monotonically decreasing, and $\gamma_i = (1-\mu_i)e_{i+}$ is monotonically increasing with the index $i$ of the attempt. In this case, the derivation is similar to that of Proposition 2. Therefore, we skip all unnecessary details and present the results directly.

**Proposition 5.** *Given the above posterior risks, the RTMP accuracy $\tilde{\rho}(n)$ for problems with a scale of $n$ is*

$$\tilde{\rho}(n) = \left( \beta_1 + \sum_{i=2}^{\infty} \beta_i \prod_{j=1}^{i-1} \alpha_i \right)^n \tag{67}$$

*Let $m$ be the width of RTBS. Let $\delta_{i|m}(n)$ denote the probability of a proposed step being rejected (either instantly or recursively) at the $i$-th attempt on a correct state, and $\epsilon_m(n)$ follows the same definition as in Proposition 2. We have $\delta_{i|m}(0) = \epsilon_m(0) = 0$ and the following recursive equations for $n > 0$:*

$$\delta_{i|m}(n) = \alpha_i + \beta_i \prod_{j=1}^{m} \delta_{j|m}(n-1) + \gamma_i(\epsilon_m(n-1))^m, \qquad i = 1, \cdots, m \tag{68}$$

$$\epsilon_m(n) = f + (1-f)(\epsilon_m(n-1))^m \tag{69}$$

*Then, the RTBS accuracy $\tilde{\rho}_m(n)$ for problems of scale $n$ is given by:*

$$\tilde{\rho}_m(n) = \prod_{t=1}^{n} \sigma_m(t), \qquad \text{where } \sigma_m(t) = \beta_1 + \sum_{j=2}^{m} \beta_j \prod_{i=1}^{j-1} \delta_{i|m}(t, m)\beta. \tag{70}$$

*In addition, $\delta_{i|m}(n)$, $\epsilon_m(n)$, and $\sigma_m(n)$ all monotonically increase and converge with respect to $n$.*

The theoretical result in this new setting becomes much more challenging to derive an exact validity condition that remains clear and understandable. However, it is still useful to derive a bound that sufficiently guarantees the effectiveness of reflection. In the following, we show that a sufficient condition becomes much stricter than that in Proposition 3.

**Proposition 6.** *Assume $\mu_1 < 1$ and $k = \inf_i \frac{\beta_{i+1}}{\beta_i}$ is the lower bound of the decay rate of the probability of accepting the correct step in multiple attempts. Then, a sufficient condition for $\tilde{\rho}(n) > \rho(n)$ is:*

$$\frac{e_{1-}}{k(1-\mu_1)} + \sup_i e_{i+} < 1 \tag{71}$$

*Proof.* Considering $\alpha_i = \mu_i e_{i-} + (1 - \mu_i)(1 - \max_i e_{i+})$, let $\underline{\alpha} = (1 - \mu_1)(1 - \sup_i e_{i+})$ be its lower bound. It can be seen that

$$\beta_1 + \alpha_1\beta_2 + \alpha_1\alpha_2\beta_3 + \cdots + \beta_m \prod_{i=1}^{m-1} \alpha_i \geq \sum_{j=1}^{m} (\underline{\alpha}k)^{m-1}\beta_1 = \beta_1 \frac{1 - (\underline{\alpha}k)^m}{1 - \underline{\alpha}k} \tag{72}$$

As $m \to \infty$, the sufficient condition for reflection validity is:

$$\frac{\beta_1}{1 - \underline{\alpha}k} > \mu_1 \tag{73}$$

$$\Longleftrightarrow \frac{1 - e_{1-}}{1 - k(1 - \mu_1)(1 - \sup_i e_{i+})} > 1 \tag{74}$$

$$\Longleftrightarrow (1 - \mu_1)(1 - \sup_i e_{i+}) > \frac{e_-}{k} \tag{75}$$

$$\Longleftrightarrow \frac{e_{1-}}{k} - \sup_i e_{i+}(1 - \mu_1) < 1 \tag{76}$$

$$\Longleftrightarrow \frac{e_{1-}}{k(1 - \mu_1)} + \sup_i e_{i+} < 1 \tag{77}$$

$\square$

# D   Implementation details

This section describes the details of the training datasets, model architectures, and hyper-parameters used in experiments. Our implementation derives the models architectures, pretraining, and SFT from LitGPT [1] (version 0.4.12) under Apache License 2.0.

## D.1   Algorithmic descriptions of reflective reasoning

Algorithms 1 and 2 presents the pseudo-code of reasoning execution through RMTP and RTBS, respectively. In practice, we introduce a reflection budget $M$ to avoid infinite iteration. If reflective reasoning fails to find a solution within $M$ steps, the algorithm retrogrades to non-reflective reasoning.

To implement RTBS, we maintain a stack to store the reversed list of parental states, allowing them to be restored if needed. Different from our theoretical analysis, our practical implementation does not limit the number of attempts on the input query $Q$ (as long as the total budget $M$ is not used up) as $Q$ has no parent (i.e. the stack is empty).

## D.2   Example CoT data

We implement predefined programs to generate examples of CoTs and self-verification. Figure 11 presents the example reasoning steps (correct) for non-reflective training and the example detailed verification for reflective training. In our practical implementations, the reasoning steps include additional tokens, such as preprocessing and formatting, to assist planning and transition. To simplify transition function $\mathcal{T}$, the example steps also include exactly how the states are supposed to be updated, which removes the task-specific prior in $\mathcal{T}$.

### D.2.1   Multiplication CoT

Each state is an expression $x_t \times y_t + r_t$, where $x_t$ and $y_t$ are the remaining values of two multiplicands, and $r_t$ is the cumulative result. For an input query $x \times y$, the expert reasoner assigns $x_1 = x$, $y_1 = y$, and $r_1 = 0$.

For each step, the reasoner plans a number $u \in \{1, \ldots, 9\}$ to eliminate in $x_t$ (or $y_t$). Specifically, it computes $\delta = u \times y_t$ or ($\delta = u \times x_t$). Next, it finds the digits in $x_t$ (or $y_t$) that are equal to $u$ and set them to 0 in $x_{t+1}$ (or $y_{t+1}$). For each digit set to 0, the reasoner cumulates $\delta \times 10^i$ to $r_t$, where $i$ is the position of the digit (starting from 0 for the unit digit). An example of a reasoning step is shown in Figure 11(a). Such steps are repeated until either $x_T$ or $y_T$ becomes 0, then the answer is $r_T$.

**Algorithm 1** Reflective reasoning through RMTP

**Require:** the query $Q$, the augmented policy $\tilde{\pi} = \{\pi, \mathcal{V}\}$, transition function $\mathcal{T}$, and reflective reasoning budget $M$

1: $t \leftarrow 0, S_t \leftarrow Q$
2: **repeat**
3:      Infer $R_{t+1} \sim \pi(\cdot \mid S_t)$
4:      $Reject \leftarrow$ False
5:      **if** $t \leq M$ **then**
6:          Infer $V_{t+1} \sim \mathcal{V}(\cdot \mid S_t, R_{t+1})$
7:          $Reject \leftarrow$ True if "×" $\in V_{t+1}$
8:      **if** $Reject =$ True **then**
9:          $S_{t+1} \leftarrow S_t$
10:     **else**
11:         $S_{t+1} \leftarrow \mathcal{T}(S_t, R_{t+1})$
12:         **if** $R_{t+1}$ is the answer **then**
13:            $T \leftarrow t + 1, A \leftarrow R_{t+1}$
14:         **else**
15:            $t \leftarrow t + 1$
16: **until** The answer $A$ is produced
17: **return** $A$

---

**Algorithm 2** Reflective trace-back search

**Require:** the query $Q$, the augmented policy $\tilde{\pi} = \{\pi, \mathcal{V}\}$, transition function $\mathcal{T}$, search width $m$, and reflective reasoning budget $M$

1: $t \leftarrow 0, S_t \leftarrow Q$
2: $i \leftarrow 0$ {The index of attempts}
3: Initialize an empty stack $L$
4: **repeat**
5:      Infer $R_{t+1} \sim \pi(\cdot \mid S_t)$
6:      $i \leftarrow i + 1$
7:      $Reject \leftarrow$ False
8:      **if** $t \leq M$ **then**
9:          Infer $V_{t+1} \sim \mathcal{V}(\cdot \mid S_t, R_{t+1})$
10:         $Reject \leftarrow$ True if "×" $\in V_{t+1}$
11:     **if** $Reject =$ True **then**
12:         **if** $i < m$ **then**
13:            $S_{t+1} \leftarrow S_t$
14:         **else**
15:            {Find a parent state that has remaining number of attempts.}
16:            **repeat**
17:               Pop $(s_k, j)$ from $L$
18:               $S_{t+1} \leftarrow s_k, i \leftarrow j$
19:            **until** $i < m$ or $L$ is empty
20:     **else**
21:         Push $(S_{t+1}, i)$ into $L$
22:         $S_{t+1} \leftarrow \mathcal{T}(S_t, R_{t+1}), i \leftarrow 0$
23:         **if** $R_{t+1}$ is the answer **then**
24:            $T \leftarrow t + 1, A \leftarrow R_{t+1}$
25:         **else**
26:            $t \leftarrow t + 1$
27: **until** the answer $A$ is produced
28: **return** $A$

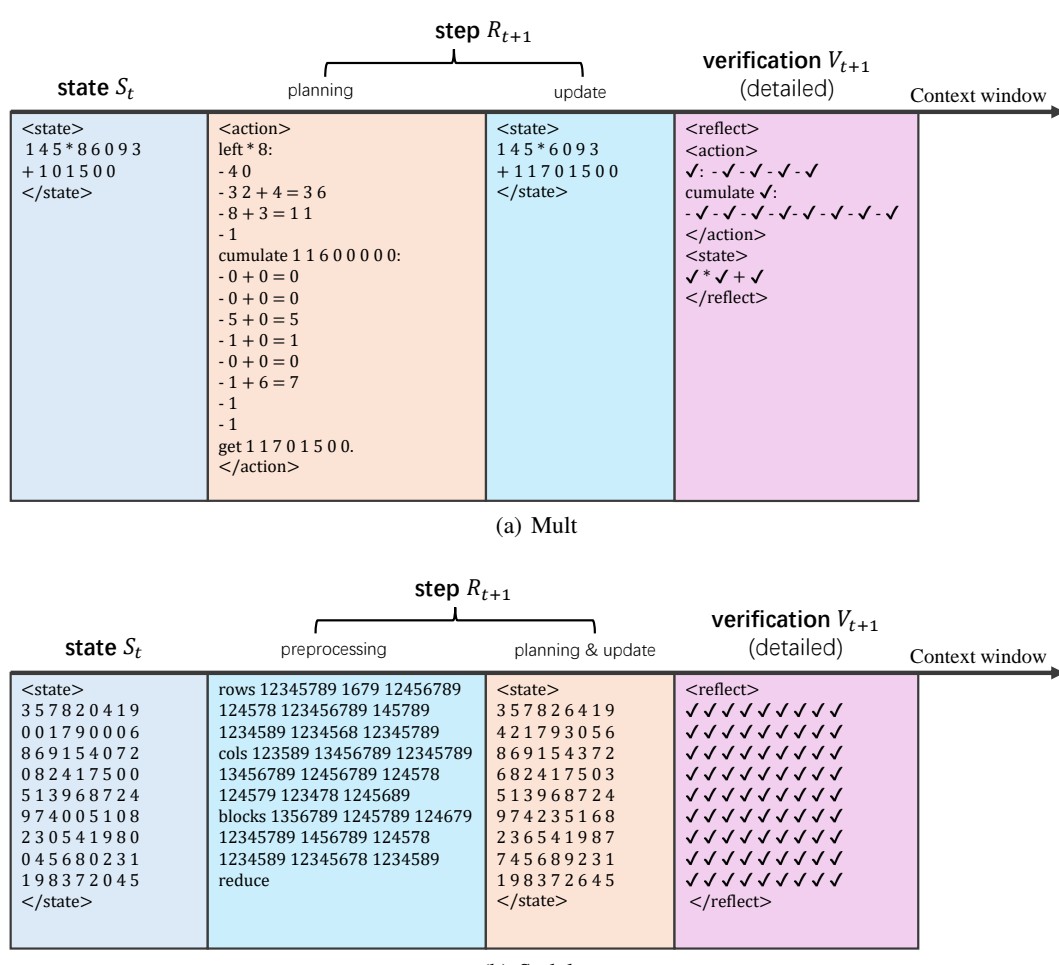

(a) Mult

(b) Sudoku

Figure 11: Example reasoning steps with detailed verfication for integer multiplication and Sudoku.

### D.2.2 Sudoku CoT

Each state is a $9 \times 9$ matrix representing the partial solution, where blank numbers are represented by $0$. On each step, the reasoner preprocesses the state by listing the determined numbers of each row, columns, and blocks. Given these information, the model reduces the blank positions that has only one valid candidate. If no blank can be reduced, the model randomly guess a blank position that has the fewest candidates. Such process continues until there exist no blanks (i.e., zeros) in the matrix.

An example of a reasoning step is shown in the right of Figure 11(b). The planned updates (i.e., which positions are filled with which numbers) is intrinsically included in a new puzzle, which is directly taken as the next state by the transition function $\mathcal{T}$.

### D.2.3 Verification of reasoning steps

**Binary Verification** The Binary verification labels are generated using a rule-based checker of each reasoning step. In Multiplication, it simply checks whether the next state $x_{t+1} \times y_{t+1} + r_{t+1}$ is equal to the current state $x_t \times y_t + r_t$. In Sudoku, it checks whether existing numbers in the old matrix are modified and whether the new matrix has duplicated numbers in any row, column, and block.

**Detailed Verification** In Multiplication, we output a label for each elemental computation — addition or unit-pair product — is computed correctly. In Sudoku, we output a label for each position in the new matrix, signifying whether the number violates the rule of Sudoku (i.e. conflicts with other

numbers in the same row, column, or block) or is inconsistent with the previous matrix. These labels are organized using a consistent format as the CoT data. Examples of detailed reflection for correct steps is in Figure 11(b). If the step contains errors, we replace the corresponding "✓" with "×".

### D.3 Model architectures and tokenization

Table 2: The model architectures of models for the transitional implementation.

| Task | Mult | | | Sudoku | | |
|------|------|------|------|--------|------|------|
| **Model size** | 1M | 4M | 16M | 1M | 4M | 16M |
| Vocabulary size | 128 | | | | | |
| Embedding size | 128 | 256 | 512 | 128 | 256 | 512 |
| Number of layers | 5 | | | | | |
| Number of attention Heads | 4 | 8 | 8 | 4 | 8 | 8 |

Our models architectures uses multi-head causal attention with LayerNorm [36] with implementation provided by LitGPT [1]. Table 2 specifies the architecture settings of transformer models with 1M, 4M, 16M parameters.

**Tokenizers**    We employ the byte-pair encoding algorithm [30] to train tokenizers on reasoning CoT examples for tiny transformers. Special tokens for reflection and reasoning structure (e.g., identifiers for the beginning and ending positions of states and reasoning steps) are manually added to the vocabulary. Since the vocabulary size is small (128 in our experiments), the learned vocabulary is limited to elemental characters and the high-frequency words for formatting.

### D.4 Hyperparameters

Table 3 presents the hyper-parameters used in training and testing the tiny-transformer models. In the following sections, we describe how these parameters are involved in our implementation.

**Non-reflective training**    The pretraining and SFT utilize a dataset $N_{CoT}$ of CoT examples generated by an expert reasoning program. Pretraining treats these CoT examples as plain text and minimizes the cross-entropy loss for next-token prediction, using the batch size $B_{pre}$ and the learning rate $\eta_{pre}$. The pretraining process terminates after predicting a total of $N_{pre\_tok}$ tokens. The non-reflective SFT uses the same dataset as that used in pretraining. It maximizes the likelihood of predicting example outputs (reasoning steps) from prompts (reasoning states), using the batch size $B_{SFT}$ and the learning rate $\eta_{SFT}$. The total number of non-reflective SFT epochs is $E_{SFT}$.

**Reflective SFT**    To perform non-reflective SFT, we use the model after non-reflective training to sample trajectories for each input query in the training set. The reflective sampling involves two decoding temperatures: the lower solving temperature $\tau_{refl:s}$ is used to walk through the solution path, while a higher proposing temperature $\tau_{refl:p}$ is used to generate diverse steps, which are fed into the reflective dataset. Then, the verification examples, which include binary or detailed labels, are generated by an expert verifier program. The reflective SFT includes $E_{RSFT}$ epochs, using the same batch size and learning rate as the non-reflective SFT.

**Reinforcement learning**    We use online RL algorithms as described in Appendix B, including PPO and GRPO. These algorithms include an experience replay buffer to store $N_{PPO:buf}$ and $N_{GRPO:buf}$ example trajectories, respectively. After every $E_{RL:int}$ epochs trained on the buffer, the buffer is updated by sampling new trajectories, using the temperature $\tau_{RL:\pi}$ for planning steps and the temperature $\tau_{RL:\tilde{\pi}}$ for reflective feedback. According to Equations 8 and 12, the hyper-parameters in both the PPO and GRPO objectives include the clipping factor $\varepsilon$ and the KL-Divergence factor $\beta$. Additionally, GRPO defines $G$ as the number of trajectories in a group. We run RL algorithms for $E_{RL}$ epochs, using the learning rate $\eta_{RL}$. PPO involves $E_{PPO:warmup}$ warm-up epochs at the beginning of training, during which only the value model is optimized.

Table 3: The main hyper-parameters used in this work.

| Task | Mult | | | Sudoku | | |
|---|---|---|---|---|---|---|
| **Model size** | 1M | 4M | 16M | 1M | 4M | 16M |
| Training CoT examples: $N_{CoT}$ | | 32K | | | 36K | |
| Total pretraining tokens: $N_{pre\_tok}$ | | | 1B | | | |
| Pretraining batch size: $B_{pre}$ | | | 128 | | | |
| Pretraining learning rate: $\eta_{pre}$ | | | $0.001 \to 0.00006$ | | | |
| SFT batch size: $B_{SFT}$ | | | 128 | | | |
| SFT learning rate: $\eta_{SFT}$ | | | $0.001 \to 0.00006$ | | | |
| Non-reflective SFT epochs: $E_{SFT}$ | | | 5 | | | |
| Reflective sampling temperature: Solving $\tau_{refl:s}$ | | | 0.75 | | | |
| Reflective sampling temperature: Proposing $\tau_{refl:p}$ | 1 | 1.25 | 1.5 | 1 | 1.25 | 1.5 |
| Reflective SFT epochs: $E_{RSFT}$ | | | 3 | | | |
| PPO replay buffer size: $N_{PPO:buf}$ | | | 512 | | | |
| GRPO replay buffer size: $N_{GRPO:buf}$ | | | 1024 | | | |
| RL sampling interval: $E_{RL:int}$ | | | 4 | | | |
| RL sampling temperature: Planning $\tau_{RL:\pi}$ | | 1.25 | | 1 | 1.25 | 1.25 |
| RL sampling temperature: Feedback $\tau_{RL:\pi_f}$ | | | 1 | | | |
| RL clipping factor: $\varepsilon$ | | | 0.1 | | | |
| RL KL-divergence factor: $\beta$ | | | 0.1 | | | |
| GRPO group size: $G$ | | | 8 | | | |
| RL total epochs: $E_{RL}$ | | | 512 | | | |
| RL learning rate: $\eta_{RL}$ | | | $0.00005 \to 0.00001$ | | | |
| PPO warm-up epochs: $E_{PPO:warmup}$ | | | 64 | | | |
| Testing first-attempt temperature: $\tau_{\pi:first}$ | | 0 | | | 1 | |
| Testing revision temperature: $\tau_{\pi:rev}$ | | | 1 | | | |
| Testing verification temperature: $\tau_{\pi_f}$ | | | 0 | | | |
| Testing non-reflective steps $T$: | | | 32 | | | |
| Testing reflective steps $\tilde{T}$: | | | 64 | | | |
| RTBS width: $m$ | | | 4 | | | |

**Testing**   During testing, we execute the reasoner using three decoding temperatures: $\tau_{\pi:first}$ for the first planning attempt, $\tau_{\pi:rev}$ for the revised planning attempt after being rejected, and $\tau_{\pi_f}$ for self-verification. We use low temperatures to improve accuracy for more deterministic decisions, such as self-verifying feedback and the first attempt in Mult. We use higher temperatures for exploratory decisions, such as planning in Sudoku and revised attempts in Mult. We set the non-reflective reasoning budget to $T$ steps and the reflective reasoning budget to $\tilde{T}$ steps. If the reflective budget is exhausted, the reasoner reverts to non-reflective reasoning. We set the search width of RTBS to $m$.

### D.5   Computational resources

Since our models are very small, it is entirely feasible to reproduce all our results on any PC (even laptops) that has a standard NVIDIA GPU. Using our hyper-parameters, the maximum GPU memory used for training the 1M, 4M, and 16M models is approximately 4GB, 12GB, and 16GB, respectively, which can be easily reduced by using smaller batch sizes. To run multiple experiments simultaneously, we utilize cloud servers with a total of 5 GPUs (one NVIDIA RTX-3090 GPU and four NVIDIA A10 GPUs).

For each model size and task, a complete pipeline (non-reflective training, reflective training, and RL) takes about two days on a single GPU. This includes 1-2 hours for non-reflective training, 8-12 hours for data collection for reflective training, 1-2 hours for reflective SFT, 6-12 hours for RL, and 6-12 hours for testing.

Table 4: The accuracy (%) of GRT-4o, DeepSeek-R1, and OpenAI o3-mini in integer multiplication and Sudoku, compared with the best performance of our 1M (1M*), 4M (4M*), and 16M (16M*) transformers. The "OOD-Hard" for LLMs only refers to the same difficulty as used in testing our tiny transformers, as OOD-Hard questions may have been seen in the training of LLMs.

| | Model | GPT-4o | o3-mini | DeepSeek-R1 | 1M* | 4M* | 16M* |
|---|---|---|---|---|---|---|---|
| | ID-Easy | 73.2 | **100** | 96.8 | 96.2 | 98.7 | 99.7 |
| Mult | ID-Hard | 32.6 | **97.2** | 77.0 | 52.7 | 77.0 | 81.1 |
| | OOD-Hard | 18.6 | **96.4** | 61.4 | 3.7 | 5.8 | 9.4 |
| | ID-Easy | 40.7 | 99.6 | 90.4 | 33.9 | 97.2 | **99.8** |
| Sudoku | ID-Hard | 2.8 | 52.8 | 4.4 | 0.4 | 58.1 | **72.2** |
| | OOD-Hard | 0.0 | 0.0 | 0.0 | 0.0 | 6.9 | **14.4** |

# E   Supplementary results of experiments

In this section, we present supplementary results from our experiments: 1) we assess the reasoning accuracy of various large language models on integer multiplication and Sudoku tasks; 2) we report the accuracy outcomes of models after implementing different supervised fine-tuning strategies; 3) we provide full results of reasoning accuracy after GRPO; 4) we additionally provide the results of PPO, which is weaker than GRPO in reflective reasoning.

## E.1   Evaluation of LLMs

In this section, we provide the reasoning accuracy of LLMs on Mult and Sudoku, including GPT-4o [22], OpenAI o3-mini [21], and DeepSeek-R1 [5]. Since GPT-4o is not a CoT-specialized model, we use the magic prompt "let's think step-by-step" [13] to elicit CoT reasoning. For o3-mini and DeepSeek-R1, we only prompt with the natural description of the queries. As shown in Table 4, among these LLMs, OpenAI o3-mini produces the highest accuracy in both tasks.

To illustrate how well tiny transformers can do in these tasks, we also present the best performance (results selected from Tables 5 and 7) of our 1M, 4M, and 16M models for each difficulty level, respectively, showing a performance close to or even better than some of the LLM reasoners. For example, according to our GRPO results (see Table 7), our best 4M Sudoku reasoner performs (RTBS through optional detailed verification) equally well to OpenAI o3-mini, and our best 16M Mult reasoner (through binary verification) outperforms DeepSeek-R1 in ID difficulties. Note that this paper mainly focuses on fundamental analysis and does not intend to compete with the general-purpose LLM reasoners, which can certainly gain better accuracy if specially trained on our tasks. Such a comparison is inherently unfair due to the massive gap in resource costs and data scale. The purpose of these results is to show how challenging these tasks can be, providing a conceptual notion of how well a tiny model can perform.

## E.2   Results of supervised fine tuning

Table 5 includes our complete results of reasoning accuracy after non-reflective and reflective SFT. As discussed in Section 3.1, our implementation uses **Reduced** states that maintain only useful information for tiny transformers. To justify this, we also test the vanilla **Complete** implementation, where each state $S_t = (Q_t, R_1 \ldots, R_{t-1})$ includes the full history of past reasoning steps. As a baseline, the **Direct** thought without intermediate steps is also presented.

**Reducing the redundancy of states in long CoTs benefits tiny transformers.** The left three columns in Table 5 compare the above thought implementations for non-reflective models. We see that both direct and complete thoughts fail to provide an acceptable performance even in ID-Easy difficulty. This proves the importance of avoiding long-context inference by reducing redundancy in representing states. Considering the huge performance gap, we exclude the complete and direct implementations from our main discussion.

**Estimated errors of self-verification**   For RMTP and RTBS executions, we employ the oracle verifiers to maintain test-time statistics of the average $e_-$ and $e_+$ (see definition in Section 4) of reasoning states. The results are shown in Table 6, where we also present the difference in how much

Table 5: The reasoning accuracy (%) for 1M, 4M, and 16M transformers after SFT.

| Thought Implementation | | | Direct | Complete | Reduced | | | | | | |
|---|---|---|---|---|---|---|---|---|---|---|---|
| Verification Type | | | None | None | None | Binary | | | Detailed | | |
| Reflective Execution | | | None | None | None | None | RMTP | RTBS | None | RMTP | RTBS |
| 1M | Mult | ID Easy | 21.8 | 10.6 | 23.6 | 95.8 | 94.5 | 93.4 | 22.0 | 33.4 | 24.2 |
| | | ID Hard | 3.0 | 1.4 | 2.0 | 52.7 | 44.6 | 35.5 | 2.2 | 4.8 | 2.8 |
| | | OOD Hard | 1.4 | 0.3 | 1.0 | 3.7 | 2.2 | 1.2 | 1.0 | 0.8 | 0.4 |
| | Sudoku | ID Easy | 2.8 | 0 | 1.4 | 33.0 | 32.4 | 2.4 | 17.4 | 18.7 | 19.4 |
| | | ID Hard | 0 | 0 | 0 | 0.3 | 0.1 | 0 | 0.1 | 0 | 0 |
| | | OOD Hard | 0 | 0 | 0 | 0 | 0 | 0 | 0 | 0 | 0 |
| 4M | Mult | ID Easy | 15.6 | 17.2 | 92.0 | 97.7 | 97.6 | 97.3 | 94.5 | 93.8 | 93.3 |
| | | ID Hard | 1.7 | 1.9 | 37.3 | 56.9 | 62.2 | 53.0 | 43.4 | 47.6 | 42.4 |
| | | OOD Hard | 1.2 | 1.0 | 2.2 | 2.9 | 1.8 | 1.1 | 3.7 | 3.3 | 2.7 |
| | Sudoku | ID Easy | 13.0 | 3.9 | 52.2 | 92.1 | 96.8 | 96.0 | 54.4 | 81.9 | 88.5 |
| | | ID Hard | 0.1 | 0 | 3.3 | 40.9 | 46.3 | 53.3 | 5.2 | 16.9 | 45.7 |
| | | OOD Hard | 0 | 0 | 0 | 0.4 | 4.0 | 6.7 | 0.0 | 1.1 | 2.0 |
| 16M | Mult | ID Easy | 15.1 | 59.2 | 99.2 | 98.8 | 98.9 | 98.8 | 99.2 | 99.5 | 98.5 |
| | | ID Hard | 1.6 | 9.6 | 65.9 | 65.2 | 76.7 | 74.9 | 65.9 | 76.4 | 73.5 |
| | | OOD Hard | 1.2 | 1.0 | 2.5 | 1.1 | 1.3 | 1.3 | 9.2 | 9.4 | 7.2 |
| | Sudoku | ID Easy | 35.8 | 15.9 | 95.7 | 97.1 | 97.9 | 92.5 | 93.0 | 99.0 | 99.7 |
| | | ID Hard | 0.4 | 0 | 48.8 | 50.1 | 53.1 | 54.8 | 46.9 | 57.9 | 70.7 |
| | | OOD Hard | 0 | 0 | 0.4 | 0.9 | 4.4 | 6.0 | 0.7 | 8.2 | 14.4 |

reflective reasoning raises the performance over non-reflective reasoning. We only count the errors in the *first attempts* on reasoning states to avoid positive bias, as the reasoner may be trapped in some state and repeat the same error for many steps.

Table 6: The percentage (%) of test-time verification errors (i.e., $e_-$ and $e_+$) after reflective SFT. Additionally, we compute $\Delta$ as the difference of how much reflective reasoning raises the performance over non-reflective reasoning, i.e. RMTP (RTBS) accuracy minus non-reflective accuracy.

| Verification Type | | | Binary | | | | | | Detailed | | | | | |
|---|---|---|---|---|---|---|---|---|---|---|---|---|---|---|
| Reflective Execution | | | RMTP | | | RTBS | | | RMTP | | | RTBS | | |
| Measurement | | | $e_+$ | $e_-$ | $\Delta$ | $e_+$ | $e_-$ | $\Delta$ | $e_+$ | $e_-$ | $\Delta$ | $e_+$ | $e_-$ | $\Delta$ |
| 1M | Mult | ID Easy | 19.3 | 4.4 | −1.3 | 14.9 | 4.9 | −2.4 | 10.2 | 18.3 | +11.4 | 24.4 | 19.4 | +2.2 |
| | | ID Hard | 3.8 | 37.6 | −8.1 | 3.6 | 33.0 | −17.2 | 0.9 | 6.9 | +2.6 | 14.5 | 7.4 | +0.6 |
| | | OOD Hard | 16.4 | 32.9 | −1.5 | 6.0 | 22.5 | −2.5 | 13.6 | 2.2 | −0.2 | 13.2 | 2.4 | −0.6 |
| | Sudoku | ID Easy | 9.9 | 35.2 | −0.6 | 31.1 | 43.9 | −30.6 | 87.1 | 0.1 | +1.3 | 85.1 | 0.1 | +2 |
| | | ID Hard | 21.1 | 31.0 | −0.2 | 33.1 | 28.6 | −0.3 | 82.8 | 0 | −0.1 | 79.4 | 0 | −0.1 |
| | | OOD Hard | 60.3 | 7.5 | 0 | 60.2 | 13.4 | 0 | 87.9 | 0 | 0 | 84.5 | 0 | 0 |
| 4M | Mult | ID Easy | 25.1 | 5.9 | −0.1 | 58.1 | 8.9 | −0.4 | 30.4 | 3.7 | −0.7 | 28.7 | 7.5 | −1.2 |
| | | ID Hard | 2.4 | 23.6 | +5.3 | 26.0 | 30.8 | −3.9 | 3.3 | 25.1 | +4.2 | 10.0 | 29.3 | −1.0 |
| | | OOD Hard | 7.5 | 42.9 | −1.1 | 18.0 | 61.7 | −1.8 | 5.9 | 28.1 | −0.4 | 10.9 | 28.2 | −1.0 |
| | Sudoku | ID Easy | 39.5 | 9.5 | +4.7 | 40.4 | 11.5 | +3.9 | 23.8 | 0.1 | +27.5 | 46.7 | 0.3 | +34.1 |
| | | ID Hard | 41.3 | 1.9 | +5.4 | 56.0 | 6.7 | +12.4 | 17.3 | 0.2 | +11.7 | 22.1 | 0.3 | +40.5 |
| | | OOD Hard | 78.5 | 0.8 | +3.6 | 70.6 | 0.6 | +6.3 | 31.5 | 0.1 | +1.1 | 35.9 | 0.1 | +2 |
| 16M | Mult | ID Easy | 11.3 | 8.6 | +0.1 | 6.1 | 9.4 | +0.0 | 15.7 | 2.1 | +0.3 | 3.8 | 2.9 | −0.7 |
| | | ID Hard | 1.4 | 13.9 | +11.5 | 1.8 | 16.9 | +9.7 | 2.5 | 7.0 | +10.5 | 4.4 | 7.2 | +7.6 |
| | | OOD Hard | 1.3 | 86.4 | +0.2 | 1.5 | 88.2 | +0.2 | 8.5 | 18.3 | +0.2 | 11.7 | 19.7 | −2 |
| | Sudoku | ID Easy | 40.1 | 3.3 | +0.8 | 10.1 | 4.7 | −4.6 | 6.6 | 1.7 | +6 | 9.1 | 6.4 | +6.7 |
| | | ID Hard | 50.5 | 4.3 | +3 | 37.2 | 9.4 | +4.7 | 15.4 | 0.1 | +11.0 | 10.6 | 0.6 | +23.8 |
| | | OOD Hard | 75.2 | 4.2 | +3.5 | 65.0 | 3.1 | +5.1 | 28.3 | 0.1 | +7.5 | 24.8 | 0.0 | +13.7 |

Our full results provide more evidence for the findings discussed in Section 5.1:

- Learning to self-verify enhances non-reflective execution for 9 out of 12 models (2 verification types, 3 model sizes, and 2 tasks), such that accuracy does not decrease in any difficulty and increases in at least one difficulty.

- RMTP improves performance over non-reflective execution for all 4M and 16M models. However, RMTP based on binary verification fails to benefit the 1M models, which suffer from a high $e_-$.

- 4M and 16M Sudoku models greatly benefit from RTBS, especially using detailed verification.

## E.3 Results of GRPO

The complete results of models after GRPO are given in Table 7. To have a convenient comparison, Table 8 presents the difference of accuracy across Table 5 and Table 7, showing that the difference of accuracy is caused by GRPO.

Table 7: The accuracy (%) of the 1M, 4M, and 16M transformers after GRPO.

| Verification Type | | | None | Binary | | | Detailed | | | Optional Detailed | | |
|---|---|---|---|---|---|---|---|---|---|---|---|---|
| Reflective Execution | | | None | None | RMTP | RTBS | None | RMTP | RTBS | None | RMTP | RTBS |
| 1M | Mult | ID Easy | 52.6 | 96.2 | 95.9 | 95.7 | 53.0 | 49.5 | 45.1 | 48.6 | 47.7 | 48.8 |
| | | ID Hard | 11.6 | 50.0 | 44.0 | 42.0 | 11.4 | 9.7 | 8.1 | 12.2 | 12.7 | 12.6 |
| | | OOD Hard | 1.1 | 2.5 | 1.9 | 1.6 | 1.0 | 0.9 | 0.4 | 1.2 | 1.3 | 1.2 |
| | Sudoku | ID Easy | 1.3 | 33.9 | 29.2 | 4.5 | 17.6 | 20.7 | 18.7 | 23.0 | 23.0 | 22.6 |
| | | ID Hard | 0 | 0.4 | 0 | 0.2 | 0 | 0.1 | 0 | 0.1 | 0.1 | 0 |
| | | OOD Hard | 0 | 0 | 0 | 0 | 0 | 0 | 0 | 0 | 0 | 0 |
| 4M | Mult | ID Easy | 98.0 | 98.6 | 98.7 | 98.8 | 98.2 | 98.0 | 98.4 | 98.2 | 98.4 | 98.6 |
| | | ID Hard | 65.6 | 73.6 | 77.0 | 76.7 | 63.0 | 64.3 | 63.2 | 63.9 | 66.8 | 66.1 |
| | | OOD Hard | 2.3 | 2.7 | 2.7 | 2.3 | 5.8 | 5.3 | 5.3 | 3.3 | 3.2 | 3.3 |
| | Sudoku | ID Easy | 58.7 | 93.8 | 97.2 | 96.7 | 57.8 | 85.3 | 92.2 | 77.0 | 94 | 98.2 |
| | | ID Hard | 3.2 | 43.9 | 53.8 | 58.1 | 5.6 | 24.7 | 47.7 | 21.4 | 37.7 | 61.3 |
| | | OOD Hard | 0 | 0.4 | 4.9 | 6.9 | 0 | 0.4 | 2.0 | 0 | 1.8 | 4.2 |
| 16M | Mult | ID Easy | 99.8 | 99.2 | 99.2 | 99.1 | 99.7 | 99.6 | 99.4 | 99.2 | 99.4 | 99.3 |
| | | ID Hard | 77.2 | 75.2 | 81.1 | 79.6 | 76.3 | 77.8 | 77.6 | 75.9 | 78.4 | 77.7 |
| | | OOD Hard | 1.8 | 1.3 | 1.8 | 1.8 | 8.4 | 8.2 | 7.4 | 6.0 | 5.5 | 5.6 |
| | Sudoku | ID Easy | 96.3 | 97.6 | 98.8 | 94.6 | 93.3 | 98.8 | 99.8 | 88.7 | 97.6 | 99.0 |
| | | ID Hard | 51.3 | 51.7 | 58.0 | 62.3 | 46.7 | 60.4 | 72.2 | 42.2 | 57.3 | 70.9 |
| | | OOD Hard | 0.7 | 0.7 | 6.0 | 7.8 | 0.2 | 6.7 | 12.0 | 0.2 | 6.7 | 11.1 |

**Reflection usually extends the limit of RL.** For reflective models, GRPO samples experience CoTs through RMTP, where self-verification $\mathcal{V}$ and the forward policy $\pi$ are jointly optimized in the form of a self-verifying policy $\tilde{\pi}$. By comparing the RMTP results (columns 3, 6, and 9) with the non-reflective model (the first column) in Table 7, we find that GRPO usually converges to higher accuracy solving ID-Hard problems in RMTP. This shows that having reflection in long CoTs extends the limit of RL, compared to only exploiting a planning policy.

Interestingly, optional detailed verification generally demonstrates higher performance after GRPO than mandatory verification. A probable explanation is that a mandatory verification may cause the reasoner to overly rely on reflection, which stagnates the learning of the planning policy.

Overall, our full results provide more evidence to better support our findings discussed in Section 5.2:

- RL enhances 24 out of 42 ID-Hard results in Table 8 by no less than 3% (measured in absolute difference). However, only 8 out of 42 OOD-Hard results are improved by no less than 1%.

- In table 9, an increase of $e_+$ is observed in 20 out of 25 cases where $e_-$ decreases by more than 5% (measured in absolute difference).

### E.3.1 The verification errors after GRPO

Furthermore, we also present the estimated errors of verification after GRPO in Table 9, in order to investigate how self-verification evolves during RL. Our main observation is that *if a model has*

Table 8: The difference of accuracy (%) of the 1M, 4M, and 16M transformers after GRPO. Positive values mean that GRPO raises the accuracy of the models above SFT.

| Reflective Training | | | None | Binary | | | Detailed | | |
|---|---|---|---|---|---|---|---|---|---|
| Reflective Execution | | | None | None | RMTP | RTBS | None | RMTP | RTBS |
| 1M | Mult | ID Easy | +29.0 | +0.4 | +1.4 | +2.3 | +31.0 | +16.1 | +20.9 |
| | | ID Hard | +9.6 | −2.7 | −0.6 | +6.5 | +9.2 | +4.9 | +5.3 |
| | | OOD Hard | +0.1 | −1.2 | −0.3 | +0.4 | 0.0 | +0.1 | 0.0 |
| | Sudoku | ID Easy | −0.1 | +0.9 | −3.2 | +2.1 | +0.2 | +2.0 | −0.7 |
| | | ID Hard | 0.0 | +0.1 | −0.1 | +0.2 | −0.1 | +0.1 | 0.0 |
| | | OOD Hard | 0.0 | 0.0 | 0.0 | 0.0 | 0.0 | 0.0 | 0.0 |
| 4M | Mult | ID Easy | +6.0 | +0.9 | +1.1 | +1.5 | +3.7 | +4.2 | +5.1 |
| | | ID Hard | +28.3 | +16.7 | +14.8 | +23.7 | +19.6 | +16.7 | +20.8 |
| | | OOD Hard | 0.1 | −0.2 | +0.9 | +1.2 | +2.1 | +2.0 | +2.6 |
| | Sudoku | ID Easy | +6.5 | +1.7 | +0.4 | +0.7 | +3.4 | +3.4 | +3.7 |
| | | ID Hard | −0.1 | +3.0 | +7.5 | +4.8 | +0.4 | +7.8 | +2.0 |
| | | OOD Hard | 0.0 | +0.4 | +4.9 | +6.9 | 0 | −0.7 | 0 |
| 16M | Mult | ID Easy | +0.6 | +0.4 | +0.3 | +0.3 | +0.5 | +0.1 | +0.9 |
| | | ID Hard | +11.3 | +10.0 | +4.4 | +4.7 | +10.4 | +1.4 | +4.1 |
| | | OOD Hard | −0.7 | +0.2 | +0.5 | +0.5 | −0.8 | −1.2 | +0.2 |
| | Sudoku | ID Easy | +0.6 | +0.5 | +0.9 | +2.1 | +0.3 | −0.2 | +0.1 |
| | | ID Hard | +2.5 | +1.6 | +4.9 | +7.5 | −0.2 | +2.5 | +1.5 |
| | | OOD Hard | +0.3 | −0.2 | +1.6 | +1.8 | −0.5 | −1.5 | −2.4 |

*a high $e_-$ before GPRO, then GRPO tends to reduce $e_-$ and also increases $e_+$.* This change in verification errors is a rather superficial (lazy) way to obtain improvements. If the model faithfully improves verification through RL, both types of errors should simultaneously decrease — such a case occurs only in the ID-Easy difficulty or when $e_-$ is already low after SFT. This highlights a potential retrograde of self-verification ability after RL.

Table 9: The percentage (%) of test-time verification errors (i.e., $e_-$ and $e_+$) after GRPO. The arrows "↑" (increase) and "↓" (decrease) present the change compared to the results in SFT (Table 6).

| Verification Type | | | Binary | | | | Detailed | | | |
|---|---|---|---|---|---|---|---|---|---|---|
| Reflective Execution | | | RMTP | | RTBS | | RMTP | | RTBS | |
| Error Type | | | $e_+$ | $e_-$ | $e_+$ | $e_-$ | $e_+$ | $e_-$ | $e_+$ | $e_-$ |
| 1M | Mult | ID Easy | 6.8↓12.5 | 3.3↑1.1 | 5.0↓9.9 | 3.5↓1.4 | 12.4↓22.6 | 17.2↓1.1 | 3.5↑27.9 | 17.6↓1.8 |
| | | ID Hard | 16.5↑20.3 | 17.6↓20.0 | 7.0↑10.6 | 13.7↓19.3 | 54.6↑55.5 | 4.6↓2.3 | 42.1↑56.6 | 5.7↓1.7 |
| | | OOD Hard | 41.2↑57.6 | 1.6↓31.3 | 40.2↑46.2 | 1.5↑24.0 | 53.9↑67.5 | 16.4↓18.6 | 57.3↑70.5 | 19.1↑21.5 |
| | Sudoku | ID Easy | 1.2↑11.1 | 1.7↓36.9 | 13.1↓18.0 | 4.0↑47.9 | 0.1↑87.2 | 0.0↓0.0 | 0.0↑87.1 | 0.4↑0.5 |
| | | ID Hard | 2.9↑24.0 | 0.9↓30.1 | 5.5↓27.6 | 0.4↑29.0 | 0.1↑83.7 | 0.0↓0.0 | 0.1↓80.3 | 0.0↓0.0 |
| | | OOD Hard | 3.1↑63.4 | 0.8↑8.3 | 4.5↓55.7 | 0.9↑14.3 | 0.5↑88.4 | 0.0↓0.0 | 0.6↑85.1 | 0.0↓0.0 |
| 4M | Mult | ID Easy | 2.5↓22.6 | 4.8↓1.1 | 30.2↓27.9 | 7.1↓1.8 | 21.1↓9.3 | 3.0↓0.7 | 5.0↓23.7 | 6.8↓0.7 |
| | | ID Hard | 53.1↑55.5 | 21.8↓1.8 | 30.6↑56.6 | 28.5↓2.3 | 20.9↑24.2 | 20.1↓5.0 | 22.0↑32.0 | 24.8↓4.5 |
| | | OOD Hard | 60.0↑67.5 | 24.3↓18.6 | 52.5↑70.5 | 40.2↓21.5 | 55.7↑61.6 | 20.9↓7.2 | 53.4↑64.3 | 22.9↓5.3 |
| | Sudoku | ID Easy | 7.9↓31.6 | 3.2↓6.3 | 2.8↑43.2 | 7.7↓3.8 | 11.5↑12.3 | 1.3↓1.4 | 27.1↑19.6 | 1.9↑2.2 |
| | | ID Hard | 28.7↑70.0 | 0.5↓1.4 | 2.2↑58.2 | 4.7↓2.0 | 4.5↑12.8 | 1.6↑1.8 | 9.2↑12.9 | 0.1↓0.2 |
| | | OOD Hard | 6.9↓85.4 | 0.7↓0.1 | 11.0↑81.6 | 0.2↓0.4 | 2.1↑29.4 | 0↓0.1 | 5.9↑30.0 | 0.1↑0.2 |
| 16M | Mult | ID Easy | 4.2↓7.1 | 7.2↓1.4 | 0.4↓5.7 | 7.8↑1.6 | 8.1↓7.6 | 1.9↓0.2 | 7.8↑11.6 | 2.7↓0.2 |
| | | ID Hard | 7.9↑9.3 | 12.8↓1.1 | 6.7↑8.5 | 15.6↓1.3 | 22.7↑25.2 | 3.9↓3.1 | 18.6↑23.0 | 4.3↓2.9 |
| | | OOD Hard | 79.0↑80.3 | 47.1↓39.3 | 89.2↑90.7 | 46.4↓41.8 | 46.0↑54.5 | 12.5↓5.8 | 46.4↑58.1 | 14.7↓5.0 |
| | Sudoku | ID Easy | 24.5↑64.6 | 6.2↑9.5 | 25.6↑35.7 | 8.5↑13.2 | 2.1↑8.7 | 0.6↓1.1 | 3.3↑5.8 | 5.4↓1.0 |
| | | ID Hard | 25.3↑75.8 | 0.0↓4.3 | 16.4↑53.6 | 2.0↓7.4 | 0.5↑15.9 | 0.0↓0.1 | 2.4↑13.0 | 0.4↑1.0 |
| | | OOD Hard | 7.9↑83.1 | 3.5↓0.7 | 12.7↑77.7 | 2.1↓1.0 | 7.8↑36.1 | 0.0↓0.1 | 6.8↑31.6 | 0.1↑0.1 |

### E.3.2 The planning correctness rate after GRPO

Table 10: The planning correctness rate ($\mu$) before and after GRPO. Each result is reported by $\mu_{\text{SFT}} \to \mu_{\text{GRPO}}$.

| Task | Verification Type | Model Size | ID Easy | ID Hard | OOD Hard |
|---|---|---|---|---|---|
| Mult | Detailed | 1M | $70.2 \to 81.7$ | $54.4 \to 59.5$ | $42.9 \to 41.9$ |
| | | 4M | $98.3 \to 99.5$ | $68.4 \to 79.1$ | $35.0 \to 38.0$ |
| | | 16M | $99.7 \to 99.9$ | $80.0 \to 85.9$ | $47.9 \to 43.4$ |
| | Binary | 1M | $98.8 \to 99.1$ | $81.2 \to 80.3$ | $42.7 \to 38.6$ |
| | | 4M | $99.3 \to 99.7$ | $77.6 \to 89.9$ | $57.1 \to 48.1$ |
| | | 16M | $99.4 \to 99.8$ | $79.6 \to 85.1$ | $75.2 \to 44.8$ |
| Sudoku | Detailed | 1M | $34.1 \to 33.0$ | $13.2 \to 12.4$ | $9.0 \to 8.6$ |
| | | 4M | $85.0 \to 86.8$ | $65.2 \to 72.0$ | $70.1 \to 70.3$ |
| | | 16M | $98.6 \to 98.1$ | $92.5 \to 94.0$ | $84.9 \to 83.9$ |
| | Binary | 1M | $59.1 \to 60.3$ | $36.6 \to 36.1$ | $19.5 \to 19.9$ |
| | | 4M | $97.3 \to 97.8$ | $80.2 \to 81.4$ | $74.5 \to 70.9$ |
| | | 16M | $99.0 \to 99.2$ | $88.5 \to 85.1$ | $68.4 \to 64.6$ |

We also report how GRPO influences the step-wise planning ability, measured by $\mu$ (defined in Section 4), across various tasks, verification types, and model sizes. Shown in Table 10, GRPO increases the planning correctness rate $\mu$ in most ID cases, except for the Sudoku models with binary verification. This indicates that the proposed steps are more likely to be correct and further reduces the overall penalties of false positive verification, making an optimistic verification bias (a high $e_+$ in exchange for a low $e_-$) even more rewarding. In particular, the planning ability shows almost no improvement in OOD problems.

### E.3.3 Reflection frequency of optional detailed verification

To show how GRPO adapts the reflection frequency for optional detailed verification, Figure 12 shows the reflection frequency of 1M and 16M transformers before and after GRPO, and the reflection frequency of the 4M model is previously shown in Section 5.2. Similarly, Figure 13 shows the reflection frequency for 1M, 4M, and 16M models in Sudoku.

According to results in Table 5, reflective execution does not improve performance for the 1M model, implying its weakness in exploring correct solutions. Therefore, GRPO does not much incentivize reflection for the 1M model. Contrarily, it greatly encourages reflection for 4M and 16M models, for they explore more effectively than the 1M model. These results align with the discussion in Section 5.2 that RL adapts the reflection frequency based on how well the proposing policy can explore higher rewards.

### E.4 Reflection frequency under controlled verification error rates

To investigate how verification error rates ($e_-$ and $e_+$) influence the reflection frequency in GRPO, we ran a controlled experiment in which the error rates were fixed by intervening with expert verifications. After each time the transformer generated a non-empty verification, we replaced the verification sequence with the expert verification, where randomized noise is injected to achieve the prescribed false-negative rate $e_-$ and false-positive rate $e_+$.

We used the 4M Mult model and ran GRPO (sampling temperature = 1.25) for 25 epochs in the in-distribution setting. We measured the fraction of steps at which the model invoked non-empty reflection ("reflection frequency") after 25 epochs. Especially, we are interested in how reflection frequency changes, given a low $e_- = 0.1$ or a high $e_- = 0.4$. In both cases, we set $e_+ = 0.1$. The results are as follows:

- Using a low $e_- = 0.1$, the reflection frequency increases to $59.8\%$ after 25 GRPO epochs.

- Using a high $e_- = 0.4$, the reflection frequency drops to $0.0\%$ after 25 GRPO epochs. That is, the model learns to completely disuse reflection.

**Discussion.** When the verifier rejects many correct steps (high $e_-$), the model learns to avoid invoking reflection, driving the observed reflection frequency to nearly $0\%$. Conversely, when $e_-$ is

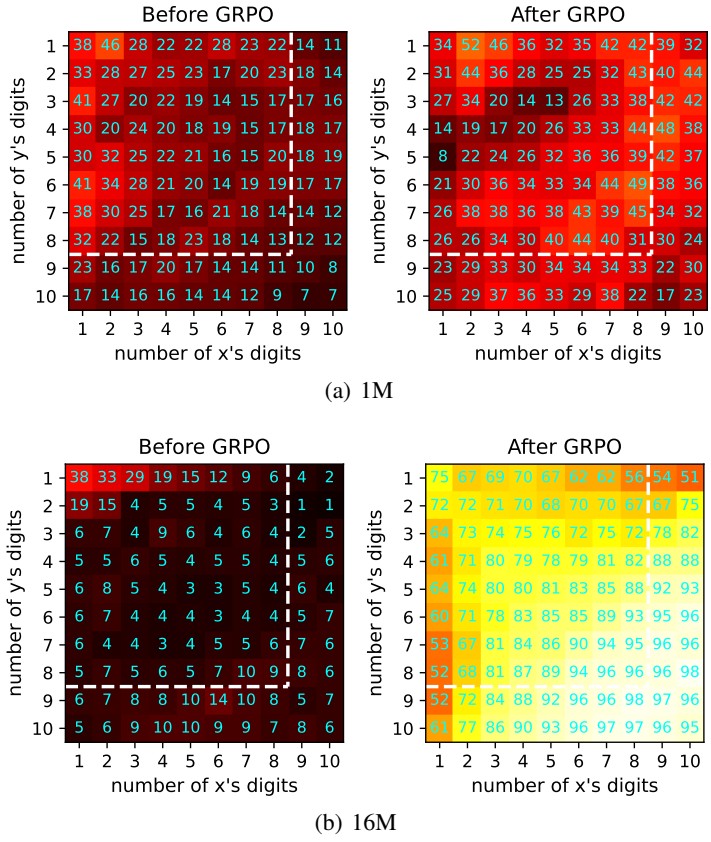

(a) 1M

(b) 16M

Figure 12: The hot-maps of reflection frequency (%) of 1M and 16M multiplication models before and after GRPO, which uses a sampling temperature of 1.25. All models are tested using RMTP execution.

low (with the same $e_+$), reflection becomes beneficial and the model increases reflection usage (here to $60\%$). Intuitively, reducing excessive false negatives shortens CoT lengths and makes reflection more rewarding; when $e_-$ is large, the model can trade off reflection for a no-reflection policy (which corresponds to the extreme $e_- = 0, e_+ = 1$), thereby avoiding costly rejections. This experiment demonstrates that the model learns to reduce $e_-$ by strategically bypassing verification.

### E.5 Results of PPO

As discussed in Appendix B.1, we prefer GRPO over PPO for tiny transformers, as the value model in PPO increases computational cost and introduces additional approximation bias in computing advantages.

Table 11 presents the reasoning accuracy after PPO, and Table 12 gives the difference compared to the SFT results in Table 5. Our results show that PPO is much weaker than GRPO. Although PPO effectively improves the non-reflective models, the performance of reflective reasoning deteriorates after PPO. To explain this, self-verification in reasoning steps causes a higher complexity of the value function, which may obfuscate tiny transformers. Overall, we suggest that GRPO is a more suitable algorithm to optimize reflective reasoning for tiny transformers.

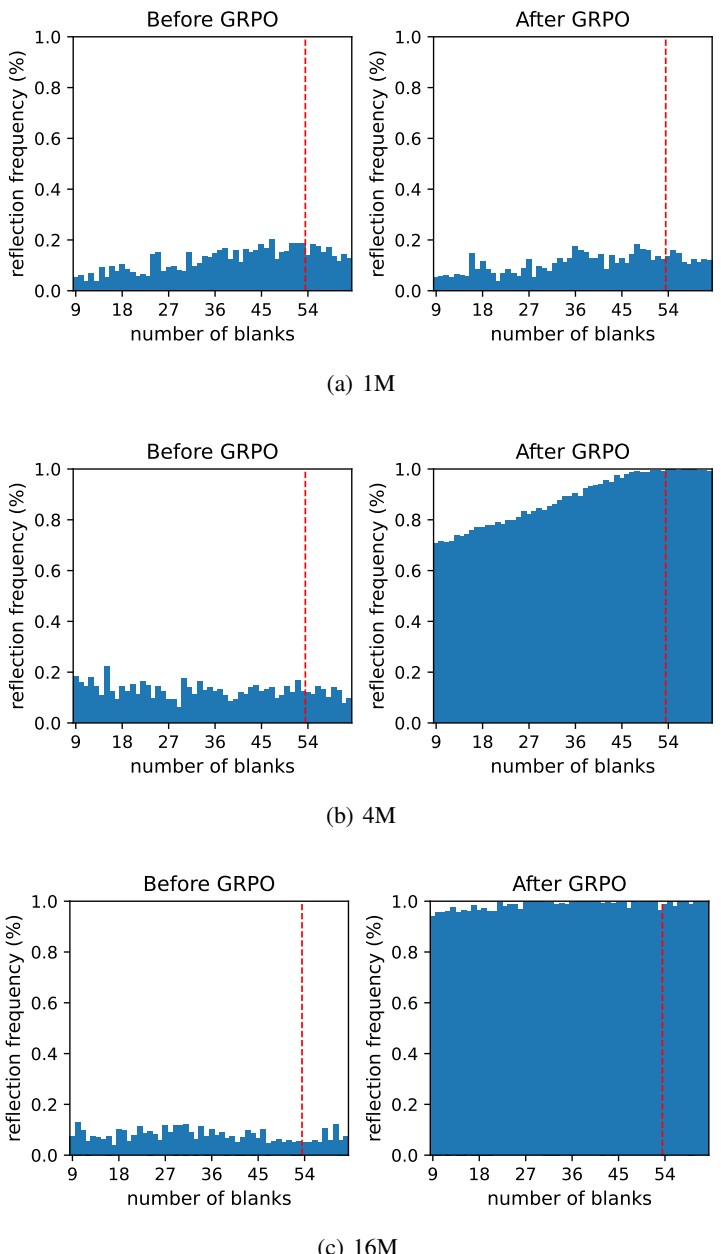

Figure 13: The histograms of reflection frequency of 1M, 4M, and 16M Sudoku models before and after GRPO, which uses a sampling temperature of 1.25. All models are tested using RMTP execution.

Table 11: The accuracy (%) of the 1M, 4M, and 16M transformers after PPO.

| Verification Type | | | None | Binary | | | Detailed | | | Optional Detailed | | |
|---|---|---|---|---|---|---|---|---|---|---|---|---|
| Reflective Execution | | | None | None | RMTP | RTBS | None | RMTP | RTBS | None | RMTP | RTBS |
| 1M | Mult | ID Easy | 39.6 | 96.5 | 94.1 | 90.6 | 28.3 | 30.1 | 27.2 | 37.9 | 49.0 | 44.4 |
| | | ID Hard | 7.8 | 49.6 | 43.7 | 32.2 | 2.4 | 3.1 | 2.4 | 5.9 | 9.6 | 7.3 |
| | | OOD Hard | 1.1 | 2.6 | 1.8 | 1.2 | 0.7 | 0.8 | 0.7 | 1.0 | 1.0 | 0.8 |
| | Sudoku | ID Easy | 1.7 | 36.1 | 33.7 | 5.6 | 17.3 | 20.6 | 20.1 | 23.8 | 21.9 | 20.1 |
| | | ID Hard | 0 | 0.4 | 1.0 | 0 | 0 | 0.1 | 0 | 0 | 0 | 0 |
| | | OOD Hard | 0 | 0 | 0 | 0 | 0 | 0 | 0 | 0 | 0 | 0 |
| 4M | Mult | ID Easy | 97.7 | 95.5 | 98.6 | 93.8 | 96.6 | 95.7 | 94.9 | 97.2 | 96.9 | 94.6 |
| | | ID Hard | 63.0 | 52.8 | 68.6 | 54.7 | 54.0 | 54.6 | 45.5 | 58.7 | 61.7 | 56.8 |
| | | OOD Hard | 2.2 | 3.1 | 2.9 | 1.6 | 5.3 | 3.9 | 2.2 | 4.4 | 3.3 | 3.7 |
| | Sudoku | ID Easy | 56.4 | 88.4 | 97.3 | 97.6 | 49.3 | 82.1 | 80.6 | 76.2 | 94.1 | 97.3 |
| | | ID Hard | 0 | 28.6 | 47.4 | 47.7 | 0 | 15.1 | 35.9 | 15.2 | 35.3 | 55.6 |
| | | OOD Hard | 0 | 0.2 | 1.6 | 3.3 | 3.1 | 0.4 | 0.9 | 0 | 1.1 | 2.7 |
| 16M | Mult | ID Easy | 99.3 | 99.0 | 99.0 | 98.2 | 98.5 | 98.7 | 97.8 | 99.0 | 99.5 | 99.2 |
| | | ID Hard | 64.8 | 62.9 | 75.7 | 71.9 | 63.2 | 68.6 | 65.6 | 65.1 | 77.1 | 74.6 |
| | | OOD Hard | 1.9 | 1.0 | 1.2 | 1.1 | 9.1 | 8.1 | 7.5 | 5.4 | 5.6 | 5.4 |
| | Sudoku | ID Easy | 96.5 | 91.8 | 97.3 | 96.7 | 87.6 | 98.1 | 98.9 | 94.5 | 96.7 | 97.1 |
| | | ID Hard | 49.0 | 41.0 | 51.4 | 52.7 | 34.7 | 55.7 | 66.3 | 47.8 | 53.8 | 53.0 |
| | | OOD Hard | 0.6 | 0 | 2.4 | 4.0 | 0 | 1.1 | 2.0 | 0 | 3.8 | 2.9 |

Table 12: The difference of accuracy (%) of the 1M, 4M, and 16M transformers after PPO. Positive values mean that PPO raises the accuracy of the models above SFT.

| Reflective Training | | | None | Binary | | | Detailed | | |
|---|---|---|---|---|---|---|---|---|---|
| Reflective Execution | | | None | None | RMTP | RTBS | None | RMTP | RTBS |
| 1M | Mult | ID Easy | +16.0 | +0.7 | −0.4 | −2.8 | +6.3 | −3.3 | +3.0 |
| | | ID Hard | +5.8 | −3.1 | −0.9 | −3.3 | +0.2 | −1.7 | −0.4 |
| | | OOD Hard | +0.1 | −1.1 | −0.4 | +0.0 | −0.3 | +0.0 | +0.3 |
| | Sudoku | ID Easy | +0.3 | +3.1 | +1.3 | +3.2 | −0.1 | +1.9 | +0.7 |
| | | ID Hard | +0.0 | +0.1 | +0.9 | +0.0 | −0.1 | +0.1 | +0.0 |
| | | OOD Hard | +0.0 | +0.0 | +0.0 | +0.0 | +0.0 | +0.0 | +0.0 |
| 4M | Mult | ID Easy | +5.7 | −2.2 | +1.0 | −3.5 | +2.1 | +1.9 | +1.6 |
| | | ID Hard | +25.7 | −4.1 | +6.4 | +1.7 | +10.6 | +7.0 | +3.1 |
| | | OOD Hard | +0.0 | +0.2 | +1.1 | +0.5 | +1.6 | +0.6 | −0.5 |
| | Sudoku | ID Easy | +4.2 | −3.7 | +0.5 | +1.6 | −5.1 | +0.2 | −7.9 |
| | | ID Hard | −3.3 | −12.3 | +1.1 | −5.6 | −5.2 | −1.8 | −9.8 |
| | | OOD Hard | +0.0 | +0.2 | +1.6 | +3.3 | +2.7 | −3.6 | −5.8 |
| 16M | Mult | ID Easy | +0.1 | +0.2 | +0.1 | −0.6 | −0.7 | −0.8 | −0.7 |
| | | ID Hard | −1.1 | −2.3 | −1.0 | −3.0 | −2.7 | −7.8 | −7.9 |
| | | OOD Hard | −0.6 | −0.1 | −0.1 | −0.2 | −0.1 | −1.3 | +0.3 |
| | Sudoku | ID Easy | +0.8 | −5.3 | −0.6 | +4.2 | −5.4 | −0.9 | −0.8 |
| | | ID Hard | +0.2 | −9.1 | −1.7 | −2.1 | −12.2 | −2.2 | −4.4 |
| | | OOD Hard | +0.2 | −0.9 | −2.0 | −2.0 | −0.7 | −7.1 | −12.4 |

