# OpenReview forum: "Self-Verifying Reflection Helps Transformers with CoT Reasoning"
_NeurIPS.cc/2025/Conference — NeurIPS 2025 poster_

### Official Review · Reviewer_NVDB · 2025-06-09

**Clarity:** 2
**Significance:** 3
**Originality:** 3
**Rating:** 4
**Confidence:** 3

**Summary:**

This paper investigates how small-scale Transformers can enhance chain-of-thought (CoT) reasoning through a “Self-Verifying Reflection” mechanism without relying on natural language. The authors propose two fundamental reflective mechanisms: Reflective MTP (RMTP) and Reflective Trace-Back Search (RTBS), which enable the model to detect and reject incorrect steps during the reasoning process. By training models with varying parameter scales on integer multiplication and Sudoku tasks, the authors demonstrate that, under reasonable constraints on verification error, self-verification can improve reasoning accuracy.

**Questions:**

1.The training pipeline depends on an “expert verifier” to label the correctness of each CoT step. Have you evaluated the accuracy of this verifier? Could its errors introduce systematic bias during training?
2.The RTBS method requires setting a backtrack width $m$. Have you conducted sensitivity analysis on the choice of $m$? During inference, how does the model determine whether backtracking is needed? Have you considered an adaptive mechanism?
3.Theoretical analysis assumes each reasoning step can be labeled as a positive or negative state. In real-world scenarios, how would you approximate the partitioning of S⁺ₙ and S⁻ₙ?

**Ethical Concerns:**

["NO or VERY MINOR ethics concerns only"]

**Final Justification:**

The authors have provided clarifications to several of our earlier concerns in the rebuttal, which helped improve the overall understanding of the paper.

**Limitations:**

yes

**Quality:**

3

**Strengths And Weaknesses:**

Strengths:
	1.The proposed “Reflective MTP” and “RTBS” frameworks integrate verification mechanisms into the reasoning sequence, allowing small models to perform efficient and minimalistic self-reflection without natural language.
	2.The chosen tasks are well-designed, aligning with the capacity boundaries of small models while highlighting the benefits of CoT structure. The experimental validation conditions are clearly defined.
	3.The introduction of a trace-back strategy for retrying previous steps significantly improves the success rate in Sudoku tasks, empirically supporting the theoretical claim that complex problems benefit from backtracking.

Weaknesses:
	1.During the Reflective SFT phase, the authors rely on an external rule-based expert verifier for supervision but do not provide evidence of the verifier’s accuracy or analyze its impact on training convergence.
	2.Although GRPO is used in the reinforcement learning phase, results show that RL primarily improves performance on in-distribution (ID) data, while verification ability degrades (e.g., increased false positive rate e⁺), suggesting that the model may be learning to bypass reflection to avoid penalties rather than improving planning strategies.
	3.While the theoretical section is mathematically rigorous, it is built on highly idealized state partitions (e.g., S⁺ₙ and S⁻ₙ), which are not observable in real tasks, limiting the practical applicability of the theorems.

---

> ### Author Rebuttal · Authors · 2025-07-25
>
> We sincerely thank the reviewer for the thoughtful and constructive feedback.
>
> ## Weakness 1 and Question 1
>
> We would like to clarify that the expert verifiers used in our experiments are effectively 100% accurate. Our tasks, such as Sudoku and integer multiplication, have well-defined and structured states. Since the models are trained from scratch on task-specific data with a constrained output format, the expert verifier can reliably parse the output and rigorously check the reasoning states against clearly defined rules.
>
> While an imperfect expert verifier would increase error rates ($e_-$ and/or $e_+$) of the learned model, the system remains functional as long as these errors are properly bounded. As discussed in the paper, perfect verification is not required for RMDP or RTBS to improve performance, so learning from an imperfect verifier is not a critical issue.
>
> ## Weakness 2
>
> We acknowledge that the observed limitation of reinforcement learning (RL) is a valid concern. However, this reflects a general challenge with RL rather than a shortcoming of our work specifically. Our intention was to explore the impact of RL on reflective reasoning, not to claim that RL substantially improves planning.
>
> In fact, reporting that RL may encourage the model to bypass reflection to avoid penalties is an important contribution of our paper, as this could also be the case of DeepSeek-R1 and other LLMs trained with RL. Further, addressing this issue and improving RL to prevent such behavior could be an important direction for future research, especially in the context of general LLM reasoning.
>
> ## Question 2
>
> The traceback is needed on a state if there are $m$ proposed steps rejected on this state. Additional theoretical insights on the choice of backtrack width $m$ are provided in Appendix C.3, Proposition 3. It shows that selecting $m$ such that $\frac{1}{\mu(1 - e_-)} < m < \frac{1}{f}$ ensures that the accuracy drop remains stable as the problem scale $n$ grows large. If $m$ falls outside this range, the success rate $\tilde{\rho}_m(n)$ tends to zero as $n$ increases.
>
> Using the recurrent formula from Proposition 1 (Appendix C.2), we have implemented a simple program to plot success rates based on the parameters $\mu$, $e_+$, $e_-$, $f$, problem scale $n$, and the RTBS width $m$. Unfortunately, we cannot include image links here, but we plan to release this code upon acceptance and add a related discussion in the revised manuscript. Our key observations are as follows:
>
> - A small $m$ causes $\tilde{\rho}_m(n)$ to decline sharply when $n$ is small. In contrast, a higher $m$ lets $\tilde{\rho}_m(n)$ drop faster as $n$ grows large.
> - The optimal $m$ to maximize $\tilde{\rho}_m(n)$ for large $n$ is the greatest integer within the interval
>
> $$\left(\frac{1}{\mu(1 - e_-)}, \frac{1}{f}\right),$$
>
> if such an integer exists.
>
> **Regarding adaptive choice of $m$:** In this work, we focus on a minimalistic design and therefore do not employ an adaptive strategy for selecting $m$. However, based on the observations above, it is possible to extend our approach by adapting $m$ according to the problem scale $n$—for example, using a larger $m$ when $n$ is small. We thank you for providing this valuable comment, and plan to add some discussion about the adaptive $m$ is the revision.
>
> ## Weakness 3 and Question 3
>
> We agree that the binary partition into $S_n^-$ and $S_n^+$ is an idealization that may not hold in all real-world scenarios. The purpose of this partition is to provide a clear and accessible theoretical framework so readers can understand our results intuitively.
>
> A more general formulation of accuracy $\rho$ using Bellman equations is discussed in Appendix C.1. However, solving for $\rho$ in this setting requires sophisticated methods such as dynamic programming, making it difficult to derive concise conclusions.
>
> To better accommodate real-world cases, future work could explore soft labeling schemes (e.g., continuous values) instead of coarse binary partitions (if possible, in the revised manuscript). For example, this soft value can be defined as *the value function under an optimal policy*. The verification ability may be defined as the expected Q function of accepted steps. Intuitively, we expect similar conclusions that bounded verification errors still enable performance gains. However, a rigour analysis would be significantly more complex. Even with our ideal setting, deriving accuracy properties under reflection is already challenging.
>
> To the best of our knowledge, no other studies have tried to given similar analysis to this self-verifying reflection for LLM reasoning. We believe that establishing a groundwork with a simple, clear model is a critical first step that can inspire further research extending these results to more advanced and realistic settings.
>
> ---
>
> We sincerely hope this clarifies the concerns raised and demonstrates the rigor and potential of our approach. Thank you again for your valuable feedback. Should you have any further questions, please feel free to raise them in our subsequent discussions.

---

> > ### Author Response · Authors · 2025-08-05
> >
> > Dear Reviewer,
> >
> > As the discussion phase draws to a close, we wanted to take a moment to reach out with sincere gratitude for your time and careful review of our manuscript. We fully understand that there may be various understandable reasons you haven’t yet had the chance to engage in the discussion. If you have any further thoughts, feedback, or perspectives on our work that you might be willing to share, we would be deeply grateful. Your insights would be valuable as we move forward with revisions.
> >
> > Best regards, Authors.

---

> > > ### Comment · Area_Chair_zqPW · 2025-08-08
> > >
> > > Dear Reviewer NVDB,
> > >
> > > The authors have updated their detailed responses to your concerns. As the Author-Reviewer discussion period comes to an end, please confirm whether your concerns were addressed, even if you previously gave a positive score.
> > >
> > > Best regards,
> > >
> > > AC

---

> > ### Comment · Reviewer_NVDB · 2025-08-09
> >
> > I would like to thank the authors for their efforts in providing the response. My concerns have been largely resolved.

---

> > > ### Author Response · Authors · 2025-08-09
> > >
> > > Dear reviewer,
> > >
> > > Thank you for your kind acknowledgment! We will ensure the discussion in our response is included in the revised manuscript.
> > > Should you feel our response is adequate, we would appreciate it if you might adjust the rating accordingly.
> > >
> > > Best regards,

---

### Official Review · Reviewer_NCEg · 2025-07-03

**Clarity:** 4
**Significance:** 2
**Originality:** 3
**Rating:** 5
**Confidence:** 3

**Summary:**

The paper proposes a framework to study the propensities of R1-style reasoning models to learn certain behaviours like self-reflection during training in easily verifiable environments like Sudoku and integer multiplication.

Within this framework, the paper theoretically shows that self-verification only helps when the FPR and TNR are below a fixed, bounded value. The authors also explore when does using a search tree for chain of thought becomes beneficial and provide a lower bound on the search depth based on the verifier's performance.

The authors empirically evaluate their framework and theoretical claims by studying how 1-16 million parameter transformers behave when trained in a similar way to reasoning models. They observe that while reflection boosts the performance of the transformers by default, RL optimisation generally modifies the verifiers to accept many actions, especially in OOD environments.

**Questions:**

I have listed down specific experiments and questions I had, along with the main strengths and weaknesses for clarity and continuity. The most important parts for me are:
1. To address why the acceptance rates increase, and provide some information about how the model's policy to suggest next steps changes.
2. To study when the model chooses to use self-reflection further. Particularly when the rejection rate is high and the planning policy is good.

**Ethical Concerns:**

["NO or VERY MINOR ethics concerns only"]

**Final Justification:**

I am keeping the same score as the preliminary review. The discussion addressed roughly all of my concerns, but it does not warrant a score improvement, as the scope/impact/usefulness is the same as before.

**Limitations:**

For most parts, yes. Addressing how likely the claims are to generalise to practical applications is required.

**Quality:**

3

**Strengths And Weaknesses:**

Strengths:
- The framework is well-built, and the claims seem theoretically sound. It seems useful to quickly explore new ways for training reasoning models.
- Very well written and easy to follow

---

Weaknesses:
- Parts of the experiments done are unlikely to generalise to Deepseek R1-style reasoning models:
    - Most reasoning models have poor models for verifying their claims effectively (as the authors noted in the paper). This usually results in 'overthinking' and exploring/rejecting many trajectories emerging during training (especially in Deepseek r1).
        - This seems to be analogous to high error rates for the verifier and a large search depth. Are there any bounds for RTBS with a poor verifier as well?
        - In contrast, in the paper, I see the verifiers oddly have a much larger acceptance rate (state-action pairs it classifies as being positive/correct) after RL. Is there any reason for this dissimilarity? Is it possible to plot the value of $\mu$ and the change in error rates?
        - Can we compare the error rates in this work with those of reasoning models using previous works like [He et al.](https://arxiv.org/pdf/2502.19361)?
    - In many practical tasks, reflection is not done after every step/regularly. Either due to it being hard (similar to why MCTS didn't work for deepseek r1), or the language model just learns this policy. Therefore, exact recursive backtracing, like in RTBS, might not be what is happening. However, this only affects a part of the motivation, and this could be potentially important for other types of CoT training.
        - Maybe adding some randomness to which states are reflected on can simulate an imperfect verifier like this.
        - Another important factor here is studying the propensity to use self-reflection when $\mu$ and $e_+$/$e_-$ change. Entire RL runs seem expensive, but artificially changing the values/keeping them fixed to see if the frequency changes after a few epochs seems reasonable.
- Slightly unclear result:
    - Moving verifiers to classify many states as positive seems like RL is trying to circumvent the verification step. The verification step still looks to be doing something useful. Maybe plotting the number and type of states pruned/rejected from this step before and after RL will make things clearer.

---

Overall: This is a technically sound work, and the framework proposed definitely has good use cases. There could be some more experiments/ablations to simulate more realistic scenarios. However, this paper still provides valuable insights.

---

> ### Author Rebuttal · Authors · 2025-07-25
>
> We thank the reviewer for carefully reading our paper and providing valuable comments.
>
> > **Are there any bounds for RTBS with a poor verifier as well?**
>
> A trivial bound indeed exists. First we notice that when $m \to +\infty$, RTBS reduces to RMDP (non-traceback), so the RMTP's bounds also hold for RTBS, given **sufficiently large** $m$. Then, for finite $m$, it is naturual that there eixsts a margin $\varepsilon > 0$ such that $\text{RMDPBound} - \varepsilon$ becomes the RTBS bound, depending on how large $m$ is. Despite the existence, the exact RTBS bound can not be written analytically for arbituary $m$ and $n$. It can only be found numerically, using our recurrent expression of accuracy $\tilde{\rho}_m(n)$ in Proposition 1 (see Appendix C).
>
> Theorem 1 shows that if $m$ is chosen within a proper range and $f > \alpha$ (depending on the task’s state representations), RTBS outperforms RMDP for the same false positive ($e_+$) and false negative ($e_-$) rates. In this case, the bounds for RMDP naturally become the bounds for RTBS, provided $n$ is sufficiently large and the task satisfies certain properties.
>
> ---
>
> > **In the paper, verifiers have a much larger acceptance rate after RL. Is there a reason for this? Can you plot the value of $\mu$ and changes in error rates?**
>
> The increased acceptance rate indicates the verifier becomes more optimistic about step correctness. As discussed in Section 4 (rigorous theory in Appendix C.4 and C.5), false positives are less harmful than false negatives. The bound for $e_-$ becomes much stricter than $e_+$ considering the posterior risks in practice. More importantly, Appendix C.4 shows that a high false negative rate blocks reward paths, *greatly increase the number of total steps to reach the answer*. As a result, a large $e_-$ block rewards of true positives, since only limited computational resrouce is allowed in practical RL.
>
> In the above case, reducing $e_-$ becomes very rewarding, as it shortens the CoT lengths so the problem can be solved within limited steps. Therefore, the reward of reducing $e_-$ surpasses the penalty of increasing $e_+$. This leads to a simple strategy to significantly increase reward gain: *an optimistic verification prior*. **This optimistic bias of verification is extremely easy to learn** --- By simply tuning a few parameters of the *output linear layer*, the transformer easily gives a high bias to the "Accept" token (meanwhile suppress the "Reject" token), whenever it needs to predict a verification token. This reduces $e_-$ and increases $e_+$, without learning any substantive new knowledge.
>
> We note that, in RL, there is no direct supervision to learn verification, and the only signal is the final reward --- If there is a lazy way to increase rewards, RL will not learn the hard way. This explains why RL tends to maximize reward by learning such an optimistic bias.
>
> **Values of $\mu$.**  We did not save runtime values of $\mu$ and verification error rates during RL training, so plotting their training curves is currently not possible without rerunning experiments. However, we can provide evaluations of $1 - \mu$ (planning error rate) before and after GRPO. Due to limited character counts, we here only present results for the multiplication task below, showing that GRPO reduces planning errors (increases $\mu$) for 4M and 16M models in ID cases:
>
> | Setting | Easy ID | Hard ID | Hard OOD |
> |---------|---------|---------|----------|
> | Mult, Detailed Verification, 4M, **Before** GRPO | 1.68 | 31.60 | 65.03 |
> | Mult, Detailed Verification, 4M, **After** GRPO | 0.46 | 20.95 | 61.99 |
> | Mult, Binary Verification, 4M, **Before** GRPO | 0.67 | 22.40 | 42.93 |
> | Mult, Binary Verification, 4M, **After** GRPO | 0.33 | 10.10 | 51.86 |
> | Mult, Detailed Verification, 16M, **Before** GRPO | 0.27 | 20.00 | 52.06 |
> | Mult, Detailed Verification, 16M, **After** GRPO | 0.13 | 14.09 | 56.62 |
> | Mult, Binary Verification, 16M, **Before** GRPO | 0.60 | 20.44 | 24.85 |
> | Mult, Binary Verification, 16M, **After** GRPO | 0.24 | 14.91 | 55.23 |
>
> The above indicates an increase in $\mu$ after GRPO means the model produces more correct steps, which further reduces the overall penalties of false positive verification. This even makes an optimistic verification bias even more rewarding --- If $\mu$ is low, then $\tilde\rho(n)$ quickly drops to nearly $0$ as $n$ increases, where this optimistic verification bias does not translate to a significant advantage. By increasing $\mu$, the advantage of optimistic verification becomes more evident for large $n$.
>
> We will add this discussion to the revised manuscript.
>
> ---
>
> > **Can error rates in this work be compared with those of reasoning models like in He et al.?**
>
> Currently, such a comparison is impractical because:
>
> 1. He et al. measure "effective reflection" differently and do not discuss error rates explicitly.
> 2. Estimating error rates with low variance requires many steps, which is expensive for LLM-based reasoners.
> 3. LLMs reason in flexible natural language, complicating verification error evaluation. Enforcing strict formats may affect their performance.
>
> ---
>
> > **Reflection is not done after every step in many practical tasks.**
>
> We agree this is important. Our "optional reflection" setting simulates irregular reflection with randomness, and we find it often outperforms mandatory reflection after GRPO (see Table 7 in the Appendix). However, due to some exceptions and limited evidence, we did not include this in the main paper.
>
> Your suggestion to study how reflection frequency varies with $e_-$, $e_+$, and $\mu$ is valuable. While artificially intervening on $\mu$ is not straightforward, we can fix $e_-$ and $e_+$ by manipulating the verifier using the expert.  After each time the transformer generates a valid (non-empty) verification, we replace the generated verification with the expert verification, injecting noise to control $e_-$ and $e_+$. In this way, we now conduct a supplementary experiment to investigate how the reflection frequency is affected by fixed values of $e_-$ and $e_+$ in GRPO. Although we cannot show figures here, we briefly summarize the results for the 4M Mult model:
>
> - With $e_- = 0.4$ and $e_+ = 0.1$, the find that the reflection frequency drops to almost $0$ on in-distribution problems after 25 GRPO epochs.
> - With $e_- = 0.1$ and $e_+ = 0.1$, the find that the reflection frequency on in-distribution problems rises to 60% after 25 GRPO epochs.
>
> It demonstrates that RL induces lower reflecton frequency if the verifier rejects many correct steps (a high $e_-$). This is reasonable, as adapting reflection frequence is an approach of trading-off between $e_-$ to $e_+$ (when the model does not reflect, it can be seen as $e_- = 0, e_+ = 1$). By having a low reflection frequency, it avoid rejection rates that are unecessarily high. This shows that reducing excessive rejection ($e_-$) in GRPO may be a crucial condition to incentivize high reflection frequency.
>
> We will incorporate this discussion into the revised manuscript. Subject to computational resources, we also plan to provide more comprehensive results on this issue in the revision.
>
> ---
>
> > **Does RL circumvent verification by classifying many states as positive? Can you plot the number and types of states pruned/rejected before and after RL?**
>
> It is indeed unclear whether a "positive" classification means the verifier ignores computation or genuinely verifies the step.
>
> Regarding rejection rates:
>
> - The rejection rate of positive reasoning is $e_-$ (already presented).
> - The rejection rate of false reasoning is $(1 - e_+)$ (where $e_+$ is already presented).
> - We also recorded overall rejection rates, which decrease after GRPO, as shown below:
>
> | Setting | Easy ID | Hard ID | Hard OOD |
> |---------|---------|---------|----------|
> | Mult, Detailed Verification, 1M, **Before** GRPO | 39.61 | 48.93 | 50.40 |
> | Mult, Detailed Verification, 1M, **After** GRPO | 27.98 | 43.59 | 51.04 |
> | Mult, Binary Verification, 1M, **Before** GRPO | 5.30 | 48.61 | 61.96 |
> | Mult, Binary Verification, 1M, **After** GRPO | 1.48 | 34.88 | 61.00 |
> | Mult, Detailed Verification, 4M, **Before** GRPO | 4.78 | 47.74 | 71.04 |
> | Mult, Detailed Verification, 4M, **After** GRPO | 0.87 | 22.96 | 49.76 |
> | Mult, Binary Verification, 4M, **Before** GRPO | 6.33 | 40.14 | 64.15 |
> | Mult, Binary Verification, 4M, **After** GRPO | 1.16 | 9.41 | 32.04 |
> | Mult, Detailed Verification, 16M, **Before** GRPO | 2.32 | 25.10 | 54.75 |
> | Mult, Detailed Verification, 16M, **After** GRPO | 0.22 | 15.64 | 44.84 |
> | Mult, Binary Verification, 16M, **Before** GRPO | 9.07 | 31.21 | 89.43 |
> | Mult, Binary Verification, 16M, **After** GRPO | 1.41 | 14.82 | 67.70 |
>
> We would appreciate clarification if the reviewer meant a different type of plot regarding rejected states.
>
> ---
>
> We sincerely hope these responses address your concerns. Should you have any further questions, please feel free to raise them in our subsequent discussions.

---

> > ### Author Response · Authors · 2025-08-05
> >
> > Dear Reviewer,
> >
> > As the discussion phase draws to a close, we wanted to take a moment to reach out with sincere gratitude for your time and careful review of our manuscript. We fully understand that there may be various understandable reasons you haven’t yet had the chance to engage in the discussion. If you have any further thoughts, feedback, or perspectives on our work that you might be willing to share, we would be deeply grateful. Your insights would be valuable as we move forward with revisions.
> >
> > Best regards, Authors.

---

> > ### Comment · Reviewer_NCEg · 2025-08-06
> >
> > Thanks for the detailed rebuttal! I am satisfied with most of the points mentioned to address my questions and would only like to mention a few (relatively less important) points below:
> >
> > (1) Regarding circumventing verifiers:
> >
> > It is definitely somewhat unclear whether the model truly learns to 'not use verifiers'. Nevertheless, I think the two pieces of evidence you shared provide a bit more clarity to the point 'Does RL truly enhance verification':
> > - Planning gets better
> > - Rejection rate decreases significantly (which you have already indirectly shown in the paper)
> > Maybe looking at the verifier's performance might help? (eg, the AUROC might either remain the same or decrease: if it decreases, it might be more than just shifting both +ve and -ve distributions by a constant)
> >
> > (2) I still feel that there should be some more discussion on the disparity between the high rejection rates seen in reasoning models and the low rejection rates here:
> > - Which regime does this belong to (wrt $\mu$, error rates)?  Is it because the planning itself is very poor, and optimistic verification bias does not translate to a significant advantage, as you mentioned in the rebuttal?
> > - Is it possible to see this in these setups? Or are these environments too simplistic that RL aggressively collapses to high/low reflection frequencies (as shown in your error rates experiment) and ends up focusing on $\mu$?
> >
> > (3) Nit: A collated table of notations would be very beneficial to the readers, as it takes some time to get familiar with the terms (or remembering them when coming back to the paper).
> >
> > I shall maintain my already positive scores.

---

> ### Author Response · Authors · 2025-08-06
>
> Thank you for your kind response. We appreciate that providing more discussion to these points will strengthen our work, and will revise the manuscript accordingly.
>
> Note: A table of notations have already been in Appendix A. We mentioned a link to this table in Section 3.1, where we introduce the basic formulations. We will try to mention it in a clearer position in the revision.

---

### Official Review · Reviewer_oXtR · 2025-07-05

**Clarity:** 3
**Significance:** 3
**Originality:** 2
**Rating:** 4
**Confidence:** 2

**Summary:**

This paper investigates how self-verifying reflection enhances chain-of-thought (CoT) reasoning in small-scale transformer models. The authors demonstrate theoretically and experimentally that even tiny transformers can achieve LLM-level performance on tasks like integer multiplication and Sudoku. The study also shows that while reinforcement learning (RL) improves in-distribution performance and encourages reflection, it tends to optimize for shallow statistical patterns rather than genuinely improving the model's ability to detect errors.

**Questions:**

The main concern, as noted in the weaknesses section, is that conclusions drawn from small-scale models on narrowly defined tasks may not generalize to larger models or more complex, real-world tasks.

**Ethical Concerns:**

["NO or VERY MINOR ethics concerns only"]

**Final Justification:**

I am largely satisfied with the submission and will maintain my score at 4. However, I would not give a higher score due to my lack of familiarity with other related work.

**Limitations:**

yes

**Quality:**

2

**Strengths And Weaknesses:**

**Strengths**
- Authors use a minimalistic reasoning framework with tiny transformers to study complex behaviors like reflective CoT, enabling controlled, low-cost analysis.
- **Comprehensive and Rigorous Evaluation**: Authors combine *formal proofs* with extensive experiments across model scales, tasks, training methods (SFT, RL), and reflection strategies (RMTP, RTBS).
- The experiments demonstrate interesting findings that self-verification improves planning even without test-time reflection, and that RL promotes reflection by shifting error types rather than enhancing verification.

**Weaknesses**
- **Limited Generalizability**: Results from small-scale, non-linguistic models may not extend to full-scale LLMs in natural language settings.

- **Narrow Task Scope**: Experiments are confined to structured tasks like integer multiplication and Sudoku, limiting relevance to broader reasoning challenges.

---

> ### Author Rebuttal · Authors · 2025-07-25
>
> We sincerely thank you for your thoughtful and constructive feedback.
>
> ## Reply to Concern 1: Lack of Large-Scale Results
>
> First, we aim to highlight that our analytical framework naturally extends to general reasoning tasks. For general LLM reasoning, a state is defined as the sequence of all prior reasoning steps:
> $$
> S_t = (R_1, R_2, \ldots, R_t)
> $$
> An LLM may not explicitly verify every step; if a step lacks explicit verification, we can default to viewing it as positively verified (accepted). Thus, our framework constitutes a minimal abstraction of natural LLM reflection. That said, a key characteristic of general LLMs is their ability to implicitly segment steps using linguistic markers (e.g., sentences or paragraphs), enabling the entire chain-of-thought (CoT) to be generated via token-wise processing. As a result, the CoT structure remains rather implicit in LLM outputs, which complicates comprehensive assessment. This motivates us to explicate the reasoning patterns in small transformers and conduct an in-depth analysis.
>
> We appreciate the reviewer’s recognition of the gap between our analysis on small transformers and practical applications involving large pretrained LLMs. We are actively pursuing follow-up work to extend these insights to billion-parameter models.
>
> Preliminary findings indicate that the flexibility of natural language can introduce noise in verification, complicating reasoning accuracy. We plan to investigate these effects thoroughly and will expand the discussion of these limitations and future directions in the revised manuscript.
>
> We also wish to clarify why we have not yet applied our methods directly to large LLMs:
>
> 1. Training and evaluating large LLMs is computationally expensive, limiting the scope of experiments and making comprehensive analysis challenging.
> 2. Pretrained LLMs generate natural language outputs that often deviate from strict formats, complicating precise, rule-based verification. In contrast, our small models are trained on controlled data with strict output formats, enabling straightforward evaluation of verification errors.
>
> Given these challenges, we focused on small transformers to establish a foundational understanding. This groundwork will guide future research on scaling these methods to large pretrained models. While some findings may not transfer directly, identifying key phenomena worthy of further investigation is a crucial first step.
>
> ## Reply to Concern 2: Narrow Task Scope
>
> We believe that the choice of structured tasks such as integer multiplication and Sudoku is appropriate and does not significantly limit the relevance of our findings. The explicit reasoning structure in these tasks enables efficient and clear analysis, and we expect our insights to generalize to broader applications.
>
> **Our minimalistic framework also applies to general natural-language reasoning.** For general LLM reasoning on unstructured tasks, a state can be defined as the sequence of all previous reasoning steps:
> $$
> S_t = (R_1, R_2, \ldots, R_t)
> $$
> These steps may be separated by paragraphs (`"\n\n"`), splitters (`"---"`), sentences, or fixed token counts. Some LLM reasoners explicitly use special tokens to delimit reasoning steps. Therefore, our reflection framework extends naturally to general reasoning tasks without explicit structured states, and the theoretical analysis remains applicable.
>
> **Why focus on explicitly structured tasks?** Structured tasks allow us to identify minimal state representations, so that in the step generation
> $$
> S_t \xrightarrow{\pi} R_{t+1}
> $$
> the model receives a concise prompt. These short prompts are more suitable for small transformers and computationally efficient. Additionally, structured tasks provide straightforward ways to verify the correctness of each step, enabling precise measurement of error types such as false positives and false negatives.
>
> We emphasize that **structured chain-of-thought reasoning implicitly exists in general LLMs**. When attending to a long prompt containing all previous steps, only a subset of tokens typically receive significant attention weights (assuming a well-trained model), while redundant information is ignored. These influential tokens effectively form a minimal state representation of the reasoning process. Thus, although our analysis uses simple tasks with explicit structures, it offers valuable insights for broader reasoning problems.
>
> In this paper, we avoid unstructured reasoning tasks because (1) they generally require large models, which are computationally expensive, and (2) verifying the correctness of each reasoning step is challenging.
>
> ---
>
> We sincerely hope these responses address your concerns and clarify the scope and contributions of our work. Should you have any further questions, please feel free to raise them in our subsequent discussions.

---

> > ### Author Response · Authors · 2025-08-05
> >
> > Dear Reviewer,
> >
> > As the discussion phase draws to a close, we wanted to take a moment to reach out with sincere gratitude for your time and careful review of our manuscript. We fully understand that there may be various understandable reasons you haven’t yet had the chance to engage in the discussion. If you have any further thoughts, feedback, or perspectives on our work that you might be willing to share, we would be deeply grateful. Your insights would be valuable as we move forward with revisions.
> >
> > Best regards, Authors.

---

> ### Comment · Reviewer_oXtR · 2025-08-06
>
> Thank you for the response. I am largely satisfied with the submission and will maintain my score as is.

---

### Official Review · Reviewer_xGjd · 2025-07-05

**Clarity:** 3
**Significance:** 3
**Originality:** 2
**Rating:** 4
**Confidence:** 3

**Summary:**

This paper investigates how LLMs use reflection and self verify the reasoning chain-of-thoughts (CoTs) which leads to empirical improvements. To analytically understand self verifying reflection process, they use a reasoning framework with small transformer models. They found that their small transformers with self-verification archives performance of LLMs in integer multiplication and Sudoko. They also theoretically show that self-verifying reflection guarantees improvements if verification errors are properly bounded.

**Questions:**

1) Do the authors have any intuition on why in Figure 6, for OOD hard problem with binary verification smaller model (1M) performs better than larger models (4M and 16M)?
2) Can the authors fine-tune larger pretrained models such as Llama or Qwen-7B to test whether the RL findings generalize at that scale?

**Ethical Concerns:**

["NO or VERY MINOR ethics concerns only"]

**Limitations:**

yes

**Quality:**

3

**Strengths And Weaknesses:**

Strengths
1) To analytically understand self verification reflection the authors use small transformers which mimics the performance of LLMs. The use of small models reduces the cost of comprehensive experiments.

2) They theoretically show that a self-verifying reflection step improves reasoning as long as the verifier’s errors are properly bounded, so even a lightweight verifier suffices, rather than relying on an expensive one.

3) They also found that a reinforcement-learning (RL) finetuning with reflection can be useful only if the reasoner’s exploration strategy covers a diverse set of candidate solutions; without sufficient exploration, the RL signal can not discover the added value of reflection.

Weaknesses
1) They did the analysis with tiny models, which shows limited generalisation ability and RL finetuning didn't improve it.

2) No large scale finetuning experiments are done on reasoning benchmarks to show whether their analysis transfer well to LLMs or not.

---

> ### Author Rebuttal · Authors · 2025-07-25
>
> Dear Reviewer,
>
> Thank you very much for your thoughtful and constructive feedback. We greatly appreciate your careful reading and insightful comments.
>
> ## Response to Weaknesses
>
> ### Weakness 1: Limited Generalization of Small Models and RL Fine-tuning
>
> We did not claim that small transformers or RL fine-tuning would have strong generalization. Since our models are trained on task-specific data rather than large-scale natural language corpora, limited generalization is expected and does not diminish the core contributions of our work. Moreover, our findings help clarify a common misconception that RL fine-tuning necessarily improves generalization.
>
> That said, we agree that generalization is an important direction. Our results already indicate that learning self-verification can modestly enhance generalization (e.g., the 16M Mult model with detailed verification). We plan to further explore methods to improve generalization in small models in future work.
>
> ### Weakness 2: Lack of Large-Scale Fine-tuning Experiments on Pretrained LLMs
>
> We acknowledge the gap between our current experiments on small transformers and practical applications involving large, pretrained LLMs. We are actively pursuing follow-up work to extend these insights to billion-parameter models.
>
> Preliminary results suggest that natural language’s flexibility can sometimes introduce noise in verification, which complicates reasoning accuracy. We intend to investigate these effects thoroughly and will expand the discussion of these limitations and future directions in the revised manuscript.
>
> We also want to clarify why we have not yet applied our methods directly to pretrained LLMs:
>
> 1. Training and evaluating large LLMs is computationally expensive, limiting the scope of experiments and making comprehensive analysis challenging.
> 2. Pretrained LLMs generate natural language outputs that often deviate from strict formats, complicating precise, rule-based verification. In contrast, our small models are trained on controlled data with strict output formats, enabling straightforward evaluation of verification errors.
>
> Given these challenges, we focused on small transformers to establish foundational understanding. This groundwork will guide future research on scaling these methods to large pretrained models. While some findings may not transfer directly, identifying key phenomena is a crucial first step.
>
> ## Response to Questions
>
> ### Question 1: Why Does the Smallest Model (1M) Outperform Larger Models (4M, 16M) on OOD Hard Problems with Binary Verification?
>
> This is indeed an intriguing and somewhat surprising result. We hypothesize that learning binary verification acts as a strong regularizer, encouraging the model to extract more effective, generalizable features. This effect is more pronounced in smaller models, where many parameters must serve both forward reasoning and verification tasks. In other words, the parameter overlap between reasoning and verification is larger in small models, so improvements in verification directly enhance reasoning generalization.
>
> In contrast, larger models have more redundant parameters, allowing verification skills to develop in parameter subspaces orthogonal to those used for forward reasoning. As a result, verification learning may not improve the reasoning parameters as effectively, leading to lower OOD performance gains. This issue is the most evident for simple binary verification form, and may be addressed with more sophisticate verifications.
>
> We will include this discussion in the revised manuscript.
>
> ### Question 2: Can Larger Pretrained Models Such as LLaMA or Qwen-7B Be Fine-tuned to Test RL Findings at Scale?
>
> Testing our findings on large pretrained models is exactly our next step. As explained in our response to Weakness 2, resource constraints and evaluation challenges prevented us from including such experiments in this paper. Although it seems straightforward to shift from tiny models to these larger models, the engineering is quite different. More importantly, our current framework focuses on analytic simplicity and may not sufficiently explore the emergent portential of LLMs in solving flexible tasks using reflection, so we suggest that it would be more meaningful to fine-tune them in another study with a more sophisticated framework.
>
> Fortunately, we are now equipped to pursue follow-up research on large-scale settings. However, developing effective and efficient evaluation methods for LLM-based reasoning and verification remains a significant challenge.
>
> We apologize for not being able to provide large-scale LLM results within this rebuttal phase but look forward to sharing such results in future work.
>
> ---
>
> We sincerely hope these responses address your concerns and clarify the scope and contributions of our work. Thank you again for your valuable feedback. Should you have any further questions, please feel free to raise them in our subsequent discussions.

---

> > ### Author Response · Authors · 2025-08-05
> >
> > Dear Reviewer,
> >
> > As the discussion phase draws to a close, we wanted to take a moment to reach out with sincere gratitude for your time and careful review of our manuscript. We fully understand that there may be various understandable reasons you haven’t yet had the chance to engage in the discussion. If you have any further thoughts, feedback, or perspectives on our work that you might be willing to share, we would be deeply grateful. Your insights would be valuable as we move forward with revisions.
> >
> > Best regards, Authors.

---

### Official Review · Reviewer_bmmN · 2025-07-06

**Clarity:** 3
**Significance:** 3
**Originality:** 3
**Rating:** 4
**Confidence:** 2

**Summary:**

This paper investigates how small transformer models, without relying on natural language, can perform self-verifying reflection to improve multi-step reasoning. The authors introduce a minimal reasoning framework and provide theoretical guarantees that reflective execution improves accuracy when verification errors are bounded. Empirical results on multiplication and Sudoku tasks demonstrate that tiny transformers (1M–16M parameters) can achieve near-LLM performance using this reflective mechanism. The work also reveals that while reinforcement learning encourages reflective behavior and improves in-distribution performance, it largely exploits statistical patterns without enhancing verification capabilities.

**Questions:**

**Questions for the Authors**

1. How does your self-verifying reflection framework perform when integrated into a pre-trained transformer model, rather than training from scratch?
2. Have you considered comparing your method to natural language-based self-reflection approaches in terms of both accuracy and inference cost?

**Ethical Concerns:**

["NO or VERY MINOR ethics concerns only"]

**Final Justification:**

I appreciate author's effort in addressing all my concerns. I would love to see this paper in the final proceedings.

**Limitations:**

Yes.

**Quality:**

3

**Strengths And Weaknesses:**

**Strengths**

1. The paper is well written and clearly organized. The figures and wording are effective, making the technical content easy to follow.
2. The authors conduct a thorough analysis of the proposed method, combining theoretical guarantees with strong experimental validation.
3. The topic is timely and important. The work provides insightful observations on self-verifying reflection, which could have significant implications for improving reasoning in large language models.

**Weaknesses**

1. It remains unclear how the proposed method performs when applied to an already pre-trained model. This is a relevant concern for practical adoption, as many real-world models are not trained from scratch under controlled settings, but are instead adapted from general-purpose, pre-trained LLMs.
2. It is unclear how this method compares to other variants of CoT reasoning. How does this minimalistic reflection framework compare to natural language-based self-reflection COT in terms of (1) accuracy and (2) inference efficiency? Such comparisons would help contextualize the advantages of the proposed approach.

---

> ### Author Rebuttal · Authors · 2025-07-25
>
> Dear Reviewer,
>
> Thank you very much for your thoughtful and constructive feedback. We appreciate your recognition of the strengths of our work and the opportunity to address your concerns.
>
> ## Regarding Weakness 1 and Question 1: Integration with Pre-trained Models
> First, we aim to highlight that our analytical framework naturally extends to general reasoning tasks. For general LLM reasoning, a state is defined as the sequence of all prior reasoning steps:
>
> $$
> S_t = (R_1, R_2, \ldots, R_t)
> $$
>
> An LLM may not explicitly verify every step; if a step lacks explicit verification, we can default to viewing it as positively verified (accepted). Thus, our framework constitutes a minimal abstraction of natural LLM reflection. That said, a key characteristic of general LLMs is their ability to implicitly segment steps using linguistic markers (e.g., sentences or paragraphs), enabling the entire chain-of-thought (CoT) to be generated via token-wise processing. As a result, the CoT structure remains rather implicit in LLM outputs, which complicates comprehensive assessment. This motivates us to explicate the reasoning patterns in small transformers and conduct an in-depth analysis.
>
> We acknowledge the important gap between our current experiments and practical applications involving general-purpose, pre-trained LLMs. We are actively pursuing follow-up work to apply insights from this study to larger models with billions of parameters, which is ongoing. An intriguing preliminary finding is that natural language does not necessarily improve verification accuracy in  reasoning tasks; its flexibility can introduce noise when representing identical states or reasoning steps. We plan to carefully investigate these implications for general LLM applications in future work and will expand the discussion of these limitations and future directions in the revised manuscript.
>
> We would also like to clarify why we have not applied our method directly to pre-trained models at this stage. As discussed in the paper, evaluating reasoning methods on pre-trained LLMs presents significant challenges:
>
> 1. While we have extensively tested various scenarios with small transformers, the computational cost of training and evaluating large LLMs restricts us to limited settings, which may not reveal comprehensive patterns.
> 2. Pre-trained LLMs’ ability to understand and generate natural language complicates precise performance evaluation, especially for verification errors. Our tiny models are trained on controlled data with strict output formats, enabling straightforward rule-based evaluation. In contrast, LLMs often produce correct but format-divergent outputs, making detailed assessment difficult.
>
> Given these challenges, we chose to focus on fundamental aspects of small transformers under constrained training to map out key phenomena. This foundational understanding will guide future research on applying these insights to large pre-trained models. While some findings may not directly transfer, identifying what to investigate is a crucial first step.
>
> ## Regarding Weakness 2 and Question 2: Comparison to Natural Language-Based Reasoning
>
> We appreciate your interest in how our minimalistic reflection framework compares to natural language-based self-reflection methods in terms of accuracy and inference efficiency. For task-specific applications, we believe our approach offers competitive accuracy without sacrificing computational efficiency:
>
> - **Accuracy:** Our experiments demonstrate that tiny transformers achieve strong performance relative to natural language-based reasoners. For example, our 4M-parameter model outperforms the o3-mini model, which uses natural language reflection, on Sudoku tasks (see Appendix E.1 for LLM results). This suggests that natural language may sometimes introduce redundancy or noise that does not enhance accuracy in specific reasoning domains. We will revise the manuscript to have a direct comparison between  these natural-language reasoners and our best-performing tiny transformers in the same table.
> - **Inference Cost:** Our minimalistic framework enables efficient inference on extremely small models that do not understand natural language. For instance, batched inference of 256 queries with a 4M-parameter model can run on a laptop with only 4GB VRAM, facilitating extensive experimentation. In contrast, natural language reasoning typically requires much larger models and greater computational resources.
>
> That said, natural language approaches also have their own advantages:
>
> - They offer greater flexibility, supporting multi-task scenarios and more casual query expressions.
> - Theoretically, natural language provides a superset of reflective behaviors, potentially enabling higher performance bounds.
> - Natural language simplifies engineering by leveraging pre-trained domain priors to decompose chain-of-thought reasoning into logical steps and states via token-by-token prediction.
> - It supports revision-based reflection by explicitly expressing verification errors, although recent work [^1] suggests that LLMs may not revise effectively in practice.
>
> We will make sure to add discussion about this comparison in the revised manuscript. However, we also emphasize that our goal is not to compete directly with natural language reasoning paradigms but to provide a minimalistic framework that captures their core reflective mechanisms in a clearer, more analyzable form. The strong empirical performance of our approach is a welcome surprise rather than the primary objective. For applications demanding higher accuracy, our framework can be extended with more sophisticated reflection mechanisms, such as a formal reasoning language, in future work.
>
> ---
>
> We sincerely appreciate your valuable comments once again. Should you have any further questions, please feel free to raise them in our subsequent discussions.
>
> [^1]: He, Y. _et al._ (2025) ‘Can Large Language Models Detect Errors in Long Chain-of-Thought Reasoning?’ arXiv. Available at: https://doi.org/10.48550/arXiv.2502.19361

---

> > ### Author Response · Authors · 2025-08-05
> >
> > Dear Reviewer,
> >
> > As the discussion phase draws to a close, we wanted to take a moment to reach out with sincere gratitude for your time and careful review of our manuscript. We fully understand that there may be various understandable reasons you haven’t yet had the chance to engage in the discussion. If you have any further thoughts, feedback, or perspectives on our work that you might be willing to share, we would be deeply grateful. Your insights would be valuable as we move forward with revisions.
> >
> > Best regards, Authors.

---

> > ### Comment · Area_Chair_zqPW · 2025-08-08
> >
> > Dear Reviewer bmmN,
> >
> > The authors have updated their detailed responses to your concerns. As the Author-Reviewer discussion period comes to an end, please confirm whether your concerns were addressed, even if you previously gave a positive score.
> >
> > Best regards,
> >
> > AC

---

### Comment · Area_Chair_zqPW · 2025-08-03
**Author-reviewer discussion period in progress**

Dear Reviewers,

Thank you for your efforts in reviewing this paper.

We are now in the author-reviewer discussion period. Given the detailed author responses, we encourage active discussion during this period. If you have not already, please read their response, acknowledge it in your review, and update your assessment as soon as possible.

If you have further questions or concerns, post them promptly so authors can respond within the discussion period.

Best regards,
AC

---

### Note · Authors · 2025-08-12

We thank all reviewers and the AC for their constructive reviews and insightful questions throughout the process.

We are grateful that the submission received _uniformly positive pre-rebuttal scores_ from all five reviewers, reflecting a shared appreciation of the manuscript’s clarity, completeness, and relevance. While many submissions saw score improvements during discussion, we are encouraged that our paper was already well-received. We remain committed to improving it further and will continue refining discussion points and supplementing results in line with reviewer feedback.

Our author-reviewer discussions have largely addressed the concerns of reviewers NCEg and NVDB. We regret that we were unable to engage further with reviewers bmmN and xGjd during the discussion period, though we hope our rebuttals were helpful and illuminating. Since no further major questions were raised during discussion, we offer a final clarification on their shared interests—raised by bmmN, oXtR, and xGjd—regarding how our findings extend to LLMs and natural language.

We agree that this is an important direction and have discussed it fairly as a limitation in the paper. However, we respectfully argue that these concerns should not overshadow the value and completeness of our current contribution:

- This work centers on foundational patterns of chain-of-thought (CoT) reasoning, abstracted from natural language to ensure analytic clarity. As affirmed by reviewer oXtR, the submission is self-contained and complete even without LLM-scale experiments.
- Demonstrating reflective CoT reasoning in small transformers is itself a significant and novel result.

Scaling to LLMs presents major challenges—both computational and methodological. We are actively pursuing follow-up research to extend these insights, though reliable evaluation remains difficult and expensive. We caution against setting an implicit norm that papers must include billion-scale experiments, which risks marginalizing fundamental ideas given the diverse conditions of researchers. Instead, we advocate for a collaborative research paradigm where low-cost exploration and large-scale validation complement each other across contributors. NeurIPS offers a valuable platform for this synergy, and we are optimistic about the insights it can foster.

We hope these reflections help clarify our position and support the AC in evaluating the merits of our submission.

---

### Decision · Program_Chairs · 2025-09-17

**Decision:**

Accept (poster)

**Comment:**

The reviewers have reached a consensus on the acceptance of this paper. Although some reviewers did not participate in the discussion, the period was effective in addressing the concerns raised, thanks to the authors’ successful responses. All reviewers acknowledge its significant contribution to the community, mainly through its novelty and a well-organized manuscript with thoughtful insights. While some reviewers noted minor weaknesses related to evaluation, this AC believes the paper’s strengths far outweigh these issues, which can be addressed in a revision. This AC concurs with the reviewers’ positive views and recommends acceptance.